# Bias-corrected and spatially disaggregated seasonal forecasts: a long-term reference forecast product for the water sector in semi-arid regions

Christof Lorenz[1], Tanja C. Portele[1], Patrick Laux[1,2], and Harald Kunstmann[1,2]

[1]Karlsruhe Institute of Technology, Institute of Meteorology and Climate Research, Kreuzeckbahnstr. 19, 82467 Garmisch-Partenkirchen, Germany
[2]Augsburg University, Institute of Geography, Alter Postweg 118, 86159 Augsburg, Germany

**Correspondence:** Christof Lorenz (Christof.Lorenz@kit.edu)

**Abstract.** Seasonal forecasts have the potential to substantially improve water management particularly in water scarce regions. However, global seasonal forecasts are usually not directly applicable as they are provided at coarse spatial resolutions of at best 36 km and suffer from model biases and drifts. In this study, we therefore apply a bias-correction and spatial-disaggregation (BCSD) approach to seasonal precipitation, temperature and radiation forecasts of the latest long-range seasonal forecasting system SEAS5 of the European Centre for Medium Range Weather Forecasts (ECMWF). As reference we use data from the ERA5-Land offline land surface re-run of the latest ECMWF reanalysis ERA5. By that, we correct for model biases and drifts and improve the spatial resolution from 36 km to 0.1°. This is exemplary performed over 4 predominately semi-arid study domains across the world, which include the river basins of the Karun (Iran), the São Francisco (Brazil), the Tekeze-Atbara and Blue Nile (Sudan, Ethiopia and Eritrea), and the Catamayo-Chira (Ecuador and Peru). Compared against ERA5-Land, the bias-corrected and spatially disaggregated forecasts have a higher spatial resolution and show reduced biases and better agreement of spatial patterns than the raw forecasts as well as remarkably reduced lead-dependent drift effects. But our analysis also shows that computing monthly averages from daily bias-corrected forecasts particularly during periods with strong temporal climate gradients or heteroscedasticity can lead to remaining biases especially in the lowest- and highest-lead forecasts. Our SEAS5 BCSD forecasts cover the whole (re-)forecast period from 1981 to 2019 and include bias-corrected and spatially disaggregated daily and monthly ensemble forecasts for precipitation, average, minimum and maximum temperature as well as for shortwave radiation from the issue date to the next 215 days and 6 months, respectively. This sums up to more than 100,000 forecasted days for each of the 25 (until the year 2016) and 51 (from the year 2017) ensemble members and each of the 5 analyzed variables. The full repository is made freely available to the public via the World Data Centre for Climate at https://doi.org/10.26050/WDCC/SaWaM_D01_SEAS5_BCSD (Domain D01, Karun Basin (Iran), Lorenz et al., 2020b), https://doi.org/10.26050/WDCC/SaWaM_D02_SEAS5_BCSD (Domain D02: São Francisco Basin (Brazil), Lorenz et al., 2020c), https://doi.org/10.26050/WDCC/SaWaM_D03_SEAS5_BCSD (Domain D03: Tekeze-Atbara and Blue Nile Basins (Ethiopia, Eritrea, Sudan), Lorenz et al., 2020d), and https://doi.org/10.26050/WDCC/SaWaM_D04_SEAS5_BCSD (Domain D04: Catamayo-Chira Basin (Ecuador, Peru), Lorenz et al., 2020a). It is currently the first publicly available daily high-resolution seasonal forecast product that covers multiple regions and variables for such a long period. It hence provides a unique test-bed for

evaluating the performance of seasonal forecasts over semi-arid regions and as driving data for hydrological, ecosystem or climate impact models. Therefore, our forecasts provide a crucial contribution for the disaster preparedness and, finally, climate proofing of the regional water management in climatically sensitive regions.

*Copyright statement.* TEXT

## 1 Introduction

Since the launch of seasonal hydrometeorological forecasts, it is widely agreed that sub-seasonal to seasonal forecasts offer the promise of improved hydrological management strategies (Rayner et al., 2005). Various studies showed high potential when such information is used for planning the harvests from subsistence farmers (Patt et al., 2005), predicting and monitoring drought conditions in data-sparse regions (Dutra et al., 2013; Yuan et al., 2011) or driving hydrological models (Thober et al., 2015), for proactive drought planning (Lemos et al., 2002), predicting heavy rainfall events (Tall et al., 2012), managing irrigated agriculture (Ritchie et al., 2008), operating hydropower (Block, 2011) or for predicting high and low river flow during El Niño (Emerton et al., 2019). Washington et al. (2006) even state that for the African continent, the adaptation to current (seasonal) climate anomalies through operational decision making may reduce vulnerability to climate change. It is hence obvious that these promising perspectives led to the establishment of many global initiatives and forecast products like the Long-Range Forecast Multi-Model Ensemble from the World Meteorological Organization (https://www.wmolc.org), the C3S Multi-Model-Ensemble from Copernicus (https://climate.copernicus.eu/seasonal-forecasts), the North American Multi-Model-Ensemble (NMME, https://www.cpc.ncep.noaa.gov/products/NMME/) and the re-calibrated forecasts from the International Research Institute for Climate and Society (https://iri.columbia.edu/our-expertise/climate/forecasts/seasonal-climate-forecasts/). On the regional scale, the Intergovernmental Authority on Development (IGAD) - Climate Prediction and Application Centre (IC-PAC) has developed operational seasonal forecasts for the IGAD region across Northeast Africa (https://www.icpac.net/seasonal-forecast/) while forecasts for South America were developed within the EURO-Brazilian Initiative for Improving South American seasonal forecasts (EUROBRISA, http://eurobrisa.cptec.inpe.br). Nevertheless, many water managers are still unaware of most sources of seasonal climate forecasts (Bolson et al., 2013) or claim that the forecasts are not reliable enough for improving the decision making (Rayner et al., 2005). Hence, if seasonal forecasts are to be used effectively, it is important that, along with science advances, an effort is made to develop, communicate and apply these forecasts appropriately (White et al., 2017). Patt and Gwata (2002) defined six constraints that currently limit the usefulness of seasonal forecasts particularly in developing countries: credibility, legitimacy, scale, cognitive capacity, procedural and institutional barriers, and available choices.

Some of these constraints are based on societal aspects. They hence have to be overcome through the adaptation of seasonal forecasts to accommodate for variations in the interpretive abilities of decision makers and other potential user groups (Hartmann et al., 2002). It is particularly the scale-constraint (which refers to the inconsistency between the global forecast

models and regional conditions) that can be addressed through post-processing techniques. Furthermore, evidence of bias, e.g., in global circulation model (GCM) and regional circulation model (RCM) precipitation data has prompted many investigators to avoid direct use of climate model precipitation outputs for hydrological climate change impact analysis (Teutschbein and Seibert, 2013). Among other factors, one can loosely categorize these biases into systematic model errors (e.g., Xue et al.,

2013; Magnusson et al., 2013) and drifts (e.g., Hermanson et al., 2018), and issues due to the coarse resolution of the global forecasts which prevent the models from properly representing local features in regions with complex orography (Manzanas et al., 2018a). For seasonal forecasts, particularly the model drifts are a crucial issue with their forecast horizon up to seven months as they lead to statistical inconsistencies between forecasts from different issue dates.

Since these shortcomings of seasonal or longer-term predictions are known for a long time, there is a range of methods and

techniques for correcting model biases and drifts as well as to improve the spatial resolution. For downscaling, we generally distinguish between dynamical and empirical-statistical approaches. While the dynamical methods using a RCM is computationally highly expensive (e.g., Manzanas et al., 2018b), empirical-statistical techniques usually require reliable reference data, which is often not available particularly in data sparse regions. Nevertheless, due to their lower computational demand and relatively simple implementation in operational systems, there have been significant developments in the empirical-statistical

correction approaches in the recent years. One of the most widely used methods is the so-called *bias-correction and spatial disaggregation* (BCSD, Wood et al., 2002) which was developed for downscaling climate model outputs to a higher spatial resolution. Since its introduction, there have been numerous adjustments and changes to the "classic" BCSD-approach. One can distinguish between parametric and non-parametric techniques (e.g., Lafon et al., 2013; Crochemore et al., 2016). Abatzoglou and Brown (2012) and Ahmed et al. (2013) reversed the order in which the forecasts are bias-corrected and spatially

disaggregated, which they refer to as SDBC. Thrasher et al. (2012) applied the BCSD to daily data, Voisin et al. (2010) used rank-based scaling factors between the forecasts and a random reference ensemble to allow for different daily precipitation patterns, and Hwang and Graham (2013) replaced the interpolation-based spatial disaggregation with a stochastic approach to preserve observed local rainfall characteristics while Vandal et al. (2019) recently combined BCSD with machine learning methods. Besides the adjustments to the *univariate* BCSD-technique, recent publications also aim at multivariate extensions

(e.g., Cannon, 2018), which allow for the joint correction of different variables (e.g. temperature and precipitation) or values at different locations. There have also been numerous studies where different downscaling approaches have been compared (Tryhorn and DeGaetano, 2011; Chen et al., 2013; Gutmann et al., 2014).

Despite all these efforts, most studies focus on selected regions and only short periods of time. Furthermore, the corrected data is usually not made available to the public. In this study, we therefore present a comprehensive dataset which contains

bias-corrected and spatially disaggregated seasonal (re-)forecasts for daily precipitation, temperature, and radiation from 1981 to 2019 for four semi-arid domains in Brazil (Rio São Francisco basin), Ecuador and Peru (Catamayo-Chira basin), Sudan and Ethiopia (basins of the Tekeze-Atbara and Blue Nile) as well as Iran (Karun basin). Our study regions are marked by a strong dependency on water, food, and energy supply from water reservoirs and, hence, on a sustainable multipurpose water resources management. All regions have been hit by several severe droughts and floods particularly during the last years (e.g., Elagib

and Elhag, 2011; Marengo et al., 2018; Martins et al., 2018). Moreover, the Blue Nile Basin, which will undergo tremendous

changes due to the construction of the Grand Ethiopian Renaissance Dam (GERD) near the Ethiopian-Sudanese boarder, has been controversially debated in the public and the scientific literature (Kidus, 2019) as the filling and operation of the GERD will change downstream flow patterns significantly (e.g., Wheeler et al., 2020). This underlines the urgent need for longer-term forecasts to mitigate the impacts of climatically extreme events and improve the regions' disaster preparedness (e.g., Tall et al., 2012) as well as improve the regional water management, especially in transboundary catchments (e.g., Gerlitz et al., 2020).

Reanalysis data of the ERA5-Land (ECMWF, 2019) re-run of the land component of ERA5 climate reanalyses (Hersbach et al., 2018) is used as reference for applying the BCSD on raw forecasts from the seasonal forecasting system SEAS5 (Johnson et al., 2019) of the European Centre for Medium-Range Weather Forecasts (ECMWF). While reanalyses have clear limitations, they still provide the most comprehensive and reliable (and sometimes only) source of hydrometeorological information in such data-sparse regions.

Our final product provides a temporally and spatially consistent high-resolution data set that can be used for assessing the skill of state-of-the-art seasonal forecasts, e.g., for drought forecasting, for driving hydrological or ecosystem models, as decision support for the regional water management, or as a comprehensive repository for teaching the dealing with state-of-the-art seasonal forecast products to water managers, decision makers, and other potential end users. It is made freely available to the public through the *World Data Center for Climate* (WDCC) which is hosted by the *German Climate Computing Center* (DKRZ) in Hamburg, Germany. Therefore, our approach and published dataset address several of the above mentioned constraints of seasonal forecasts, and hence provide a significant contribution towards improving the usefulness of such information and praxis-transfer particularly in developing countries. It therefore marks a large step towards a more sustainable and timely planning of the regional water management and, hence, the adaptation to a changing climate.

## 2  Data and study areas

### 2.1  Reference data

Daily reference precipitation, average, minimum, and maximum temperature at 2 m and surface solar radiation at a high spatial resolution of $0.1°$ is obtained from the ERA5-Land offline land surface re-run of ECMWF's latest reanalysis product ERA5.

ERA5 is currently produced within the Copernicus Climate Change Service (C3S) and is the successor of the older ERA Interim reanalysis, which has been extensively used in numerous hydrological and hydrometeorological studies (e.g., Lorenz and Kunstmann, 2012; Lorenz et al., 2014, 2018). In contrast to ERA Interim, ERA5 is based on the Integrated Forecasting System (IFS) Cycle 41r2 which is run at a higher resolution of 31 km and is planned to cover the whole period from 1950 to 5 days before the present, which allows its usage in near real-time and operational applications. It has been reported in numerous studies that the performance of ERA5 is superior compared to ERA Interim (Albergel et al., 2018; Urraca et al., 2018; Mahto and Mishra, 2019). Besides improvements in the underlying model systems, this can also be attributed to the huge number of assimilated *in situ*, satellite and snow observations. ECMWF states that this number has increased from approximately 0.75 million per day on average in 1979 to around 24 million per day until 2018 (Hersbach et al., 2019).

ERA5-Land uses atmospheric forcing from the ERA5 reanalysis to consistently estimate hourly land surface variables at an enhanced spatial resolution of 9 km. While no observations are directly assimilated during the production of ERA5-Land, the millions of observations, that are used for constraining the atmospheric forcing data from ERA5, have an indirect influence on the estimated land surface parameters. Furthermore, air temperature, air humidity and pressure are corrected to account for the altitude difference between the spatial resolution of the grids of ERA5 and ERA5-Land, respectively (ECMWF, 2019, 2020).

## 2.2 SEAS5 seasonal forecasts

The fifth generation of ECMWF's seasonal forecasting system is operational since November 2017. The modelled variables are provided on a reduced O320 Gaussian grid, which corresponds to a spatial resolution of approximately 36 km. SEAS5 covers the period from 1981 to the present with forecasts issued on the first of each month. During the re-forecast period from 1981 till 2016, ECMWF provides 25 ensemble members, while this number increased to a total of 51 ensemble members in 2017. The seasonal forecasts are initialized with atmospheric conditions from ERA Interim until 2016 and the ECMWF Operational Analysis since 2017. Highlights of SEAS5 compared to previous versions include a marked improvement in sea surface temperature drift, especially in the tropical Pacific, and improvements in the prediction skill of Arctic sea ice (Haiden et al., 2018).

With its release in 2017, so far only a limited number of studies exists discussing the performance of SEAS5. For a case study over Indonesia, Ratri et al. (2019) report that after bias-correcting the seasonal forecasts towards the Southeast Asia observations (SA-OBS van den Besselaar et al., 2017) gridded rainfall product, predominantly positive predictive skill during the first two forecast months is achieved. Recently, Gubler et al. (2019) did a comprehensive performance analysis of bias-corrected SEAS5-forecasts against homogenized station data across South America. They found that, in general, prediction skill of temperature forecasts is higher than the skill of precipitation forecasts and that particularly regions which are influenced by Niño-3.4 show higher skills. Highest prediction performance can be observed, amongst some other areas, over the highlands of Ecuador and the northeastern part of Brazil. This result is beneficial for our forecast product as these South American regions include two of our study domains.

## 2.3 The four semi-arid study areas

We apply the bias-correction and spatial disaggregation of the global seasonal forecasts over four domains of different size and orographic complexity which contain five semi-arid river basins: the Karun Basin (Domain D01, Iran), the Extended São Francisco Basin (D02, Brazil), the Tekeze-Atbara and Blue-Nile Basins (D03, Ethiopia and Sudan) and the Catamayo-Chira-basin (D04, Ecuador and Peru). Main characteristics and the location of the domains and basins are shown in Table 1 and Fig. 1, respectively. It should be noted that we label the domains with numbers from D01 to D04 while the basins are labeled with two-letter abbreviations (see Table 1). This allows us to easily add further domains and basins in the future.

All domains and basins are characterized by a semi-arid climate with an extended dry period and one rainy season. The headwaters are located in mountainous areas and exhibit relatively high seasonal precipitation amounts (e.g., due to convective effects), while the downstream conditions are mainly arid. The Karun river has its source in Zard-Kuh mountain with an altitude

**Table 1.** Overview and basic characteristics of the five river basins with climate data from ERA5-Land. The climatic variability during the reference period from 1981 to 2016 is provided as the standard deviation of the annual averages. In addition to the yearly mean temperature, the yearly temperature range is given by the monthly minimum and maximum temperature in brackets. The main four month of the rainy season are provided and the respective seasonal precipitation is given as a share of the total annual precipitation. In brackets, the percentage of precipitation of the 6-month rainy season (one month prior to and after the main 4 months) is also provided.

| | Karun KA (D01) | Ext. São Francisco SF (D02) | Blue Nile BN (D03) | Tekeze-Atbara TA (D03) | Catamayo-Chira CC (D04) |
|---|---|---|---|---|---|
| Area [km$^2$] | 67313 | 740820 | 308197 | 205097 | 17761 |
| Annual rainfall [mm] | 640±128 | 858±196 | 1336±132 | 727±95 | 1666±399 |
| Mean Temperature [K] | 289±0.8 | 298±0.5 | 297±0.5 | 298±0.5 | 293±0.4 |
| Min. Temperature [K] | 265±0.5 | 289±0.9 | 287±0.8 | 286±0.8 | 287±0.6 |
| Max. Temperature [K] | 310±1.5 | 307±0.5 | 308±0.6 | 310±0.7 | 301±0.6 |
| Rainy season | DJFM | DJFM | JJAS | JJAS | JFMA |
| Seasonal precipitation [%] | 73 (94) | 60 (85) | 73 (90) | 83 (92) | 65 (81) |

of more than 4000 m, which is located in the Zagros mountains in the South-Western part of Iran. Together with its multiple tributaries, it is the main source of water for irrigation, hydropower generation and drinking water supply for the Khuzestan province and its capital Ahvaz (Bakhsipoor et al., 2019) with a population of more than 1 Mio. The much longer Rio São Francisco originates in the Canastra mountain range in the state of Minas Gerais and enters, after more than 3000 km, the

Atlantic Ocean. In order to transfer water to the water-scarce states of Ceará, Pernambuco, Paraíba and Rio Grande do Norte in the Brazilian North-East, it was decided by the Brazilian government to conduct a water division project and extend the natural basin of the Rio São Francisco (Machado, 2008; de Andrade et al., 2011; Bouckaert et al., 2020). This Extended São Francisco Basin has a drainage area of more than 700000 km$^2$ and the water is heavily used for irrigation and hydropower generation. The sources for both the Blue Nile and the Tekeze-Atbara are located in the Ethiopian Highlands with altitudes

of more than 4000 m. After they pass the Ethiopian-Sudanese boarder, they flow through mainly flat and dry areas. The Blue Nile joins the White Nile in the Sudanese capital Khartoum while the Tekeze-Atbara enters the main Nile near the city of Atbara. Together, the Blue Nile and Tekeze-Atbara deliver more than 80 % of the mean annual discharge of the main Nile (e.g., Ahmed Mordos et al., 2020), which underlines the importance of these two tributaries for Ethiopia, but also the downstream countries of Sudan and Egypt. The Catamayo-Chira has its source in the Andes at an altitude of more than 3000[m . After it

passes the Ecuadorian-Peruvian boarder, it enters the Poechos Reservoir, which is mainly used for water storage, irrigation, hydropower generation and flow regulation.

While the average temperatures do not change substantially from year to year across all study domains, standard deviations of up to 25 % of the total annual rainfall (e.g., in the Catamayo-Chira basin) indicate a highly variable amount of incoming freshwater resources, which underlines the necessity for longer-term forecasts. Moreover, particularly in the Karun and

Catamayo-Chira basins, there is a very strong climatic and elevation gradient from the head- to the tailwater within only a few

hundred kilometers. For obtaining realistic estimates of precipitation and temperatures for these mountainous headwaters, we hence need models and datasets with a reasonable spatial resolution capable of describing the smaller-scale climate variability dynamics in such complex terrain. Furthermore, the basins are heavily managed including many reservoirs which are used for maintaining water security and also electrical power supply throughout the year (e.g., von Sperling, 2012; Ahmed Mordos

et al., 2020), and all basins suffer from a lack of continuous *in situ* observations.

This combination of dependency from incoming water resources and lack of observations is of particular concern in the context of climate change. Almost all of our study regions have been hit by severe extreme events during the recent years and are assumed to experience an increase in the frequency and severity of droughts and floods in the coming years (e.g., Marengo et al., 2012; Torres et al., 2017; Andrade et al., 2021). For southern Iran, Vaghefi et al. (2019) project a climate of extended

dry periods interrupted by intermittent heavy rainfalls, which is a recipe for increasing the chances of floods. Accordingly, the first months of the rainy season 2017/2018 had the lowest ever recorded amounts of precipitation, which then led to water shortages and even societal unrest during the coming months. Only one year later, exceptionally heavy rainfall events during March and April 2019 caused severe flooding in at least 26 of Iran's 31 provinces. Similarly, the Northeast Brazil region, which also includes the Rio São Francisco Basin, suffered from a prolonged drought period from 2012 to 2016 (Martins et al., 2018)

or, according to Marengo et al. (2018), even from 2010 to 2016. Elagib and Elhag (2011) report that there has been a drastic increase in temperatures over the Sudan in line with a significant decline of rainfall over the northern half of the country. Masih et al. (2014) further state that there is a clear need for increased and integrated efforts in drought mitigation to reduce the negative impacts of droughts anticipated in the future across the African continent. Finally, Domínguez-Castro et al. (2018) analyzed wet and dry extremes in Ecuador and reported that droughts have intensified in frequency and length since the middle

of the 20th century.

Besides these climatically challenging conditions, the Tekeze-Atbara, Blue Nile and Catamayo-Chira basins are *transboundary* river basins. The headwaters of the Tekeze-Atbara and Blue Nile are located in Ethiopia and contribute, together with the Baro-Akobo (Sabot) River, about 85% of the Nile Water (Yitayew and Melesse, 2011). Both rivers cross the Ethiopian-Sudanese boarder after several hundred kilometers. The Blue and White (which comes from the South) Nile merge in the

Sudanese capital Khartoum while the Tekeze-Atbara enters the main Nile near the city of Atbara. In both Ethiopia and Sudan, reservoirs of the two rivers exist or are currently under construction. Similarly, the Catamayo-Chira originates in Ecuador, is dammed in the Poechos reservoir right after the Ecuadorian-Peruvian boarder, and finally flows into the Pacific Ocean. While the Catamayo-Chira Basin is jointly managed by Ecuadorian and Peruvian authorities, coordinated management of water-related infrastructure across the international borders of the Blue Nile and Tekeze-Atbara basins rarely exists (Wheeler

et al., 2018). A recent study has analyzed the potential of a joint transboundary water management for hydro-economic sectors particularly through the integration of the newly build Grand Ethiopian Renaissance Dam (GERD) (Digna et al., 2018).

Hence, the four study domains not only provide an optimal test-bed for the performance of seasonal forecasts in semi-arid regions, but also mark regions for which a sustainable regional water management is crucial, particularly due to the increase in the frequency and severity of climatic extreme events and the transboundary challenges.

## 3 Methods

The bias-correction and spatial disaggregation approach (BCSD, Wood et al., 2002) is a widely-used two-step technique for calibrating of, e.g., climate forecasts towards any kind of reference data. First, a quantile-mapping approach is used for matching the statistical distribution of the forecasts to the reference data at the coarse forecast-resolution. The coarse-scale climatology of the reference data is then removed from the bias-corrected forecasts. The remaining *anomalies* are then interpolated to a higher-resolution grid. Finally, the high-resolution climatology is added back to the interpolated anomalies to obtain a bias-corrected and spatially-disaggregated forecast. The BCSD approach has demonstrated its potential for improving particularly climate predictions and is hence still used in many recent studies (e.g., Thrasher et al., 2013; Ning et al., 2015; Briley et al., 2017; Nyaupane et al., 2018). However, we found that particularly the spatial disaggregation, which is often similar to a simple bias correction via linear scaling including interpolation to a higher-resolved grid can lead to unrealistic values. The disaggregation of precipitation and radiation is based on a multiplicative scaling factor, which is simply the ratio between the climatology from the forecasts and the reference data. During dry months with average precipitation values close to zero, this scaling factor can become unreasonably large (especially if there are large discrepancies between the climatologies from the forecasts and the reference), and can therefore cause unreasonable, corrected values.

Consequently, to avoid the calculation of scaling factors, we also reversed the order of bias-correction and spatial disaggregation as in Abatzoglou and Brown (2012). For the spatial disaggregation step, we apply a simple bilinear interpolation of the full precipitation, temperature, and radiation forecasts. The spatially disaggregated (or interpolated) full fields are then bias-corrected using a quantile mapping approach. However, for our final product, we still stick to the technical term BCSD, as introduced in Wood et al. (2002).

The different steps are depicted in appendix A. Here, we only summarize the key characteristics of our BCSD-implementation:

- Spatial disaggregation from the coarse SEAS5- to the higher resolved ERA5-Land grid is achieved by applying a bilinear interpolation technique to the *full* precipitation, temperature and radiation forecasts and not, as in most other studies, to the anomalies.

- The CDFs for the (re-)forecasts and reference data are based on daily data from the period 1981 to 2016.

- To estimate the CDF of the reference data and (re-)forecasts, we apply a $\pm 15$ day window around the forecasted day, respectively.

- The seasonality is taken into account by estimating the forecast distributions with forecasts from the same issue date only.

- To avoid inconsistencies in the temperature data, we correct the deviations from the mean daily temperature instead of the full maximum and minimum temperatures. After bias-correction, maximum and minimum temperature are restored by adding and removing the corrected deviations from the corrected mean temperature, respectively.

– Forecast values above or below the maximum and minimum reference quantile are corrected using the *constant correction technique* from Boé et al. (2007).

– Precipitation intermittency is corrected using the method by Voisin et al. (2010) to ensure the agreement of the wet- and dry-day frequencies from ERA5-Land and SEAS5 BCSD.

5   While several studies report that using parametric distributions can lead to more stable results (e.g., Lafon et al., 2013), we prefer to use an empirical distribution as we a) have a fairly large number of samples for both the reference and forecast CDFs and b) do not want to force precipitation, temperature, and radiation to follow a certain fixed parametric distribution. A drawback of empirical distributions, however, is the need for extrapolation when a forecasted value is beyond the maximum or minimum values from the reference period. While this is crucial for climate projections where we are interested in the occurrence of, e.g., temperatures beyond the current maximum values, we do not assume such extremely high or low values in current seasonal forecasts. If, however, a forecast contains a value beyond the maximum or minimum reference quantile, we apply the constant correction method from Boé et al. (2007).

The usage of moving windows for estimating the distributions requires special attention. This step is necessary to obtain a reasonable sample size for the reference data and allowing some climatic variability during the calibration period. However, such a moving window can lead to blurred distributions particularly during pronounced transition phases, e.g., from dry to wet, wet to dry, cold to warm, or warm to cold seasons. If the onset and end of the rainy season is well known and less variable throughout the years, it might be more appropriate to adapt the moving window to such climatic conditions. However, in this study, for our approach to be as general as possible, we are using a window with constant length.

## 4   Results

20   To assess the performance of the bias-corrected and spatial disaggregated SEAS5, we compare the seasonal forecasts before and after applying the BCSD against the reference data from ERA5-Land. For a better understanding of the impact of the correction, we separate the results according to model biases, lead-dependent effects, topographic and resolution dependent effects, and overall performance. That being said, it is difficult to separate these effects completely. As an example, the low spatial resolution of the global data can result in different amounts of convective and large-scale precipitation compared to higher-resolved reference data. The result is a bias between the forecasts and the reference data due to the different spatial resolutions and the resulting description of precipitation. It is therefore not the scope of this study to discuss all details of differences between seasonal forecasts and reference data. This holds also true for the detailed discussion of results across all five variables and four domains. We focus on selected results which should show exemplary the differences between SEAS5 and ERA5-Land and how the BCSD is able to correct for these.

## 4.1 Model biases

The most obvious effect of the bias-correction is the correction of systematic model biases in the raw forecasts. Fig. 2 shows the bias of area-averaged SEAS5 forecasts before and after applying the BCSD for the five study basins. First, no simple over or underestimation of SEAS5 with respect to ERA5-Land can be observed. During certain months, the precipitation, temperature and radiation biases can reach values of up to $\pm 5$ mm/day (CC), $\pm 2$ K (KA) and $\pm 30$ W/m$^2$ (TA and BN), respectively. The biases show strong annual cycles for most basins and variables. Precipitation biases reach peak values during the main months of the rainy season (KA, SF, TA, BN, CC) and also during the transition from the dry to the rainy season (SF).

The temperature biases show much more complex patterns. While the SF- and CC-basins show an annual cycle in the temperature forecasts, particularly the forecasts for the TA- and BN-basins also reveal highly lead dependent effects. As an example, over the TA-basin, the lead-0 and lead-2 forecasts for July (i.e. the forecasts that have been issued in July and May, respectively) show biases of about -1.3 K and -0.4 K, respectively. Such large gaps between the temperature forecasts from different issue dates can also be observed over the KA- and BN-basins.

For the KA-basin, the biases of average and minimum temperature from SEAS5 reach peak values during the main months of the rainy season (around January), while the biases of maximum temperature show an opposite behaviour with maximum biases during the dry season. Over the SF-Basin, the biases of mean and maximum temperature both show a tendency towards positive biases during the dry season. Minimum temperature, however, is generally underestimated in the raw SEAS5 forecasts of SF-Basin, which leads to a negative bias throughout the year.

The biases of mean and minimum temperature forecasts over the CC-Basin also show an annual cycle with the maximum deviations from ERA5-Land around April, which also marks the end of the rainy season. But in contrast to the SF-Basin, the biases of mean and minimum temperatures show a very similar cycle while there is a positive bias of maximum temperature almost throughout the year in CC-Basin.

The bias from the radiation forecasts also shows an annual cycle with peak values either at the transition from the dry to the wet season (SF) or during the main months of the rainy season (KA, SF, TA, BN, CC).

All these effects are almost completely removed after applying the BCSD. The bias-corrected forecasts do not show any systematic positive or negative biases when compared against ERA5-Land. However, while the biases of the BCSD temperature forecasts for lead-1 to lead-6 for the KA-basin are reduced compared to the raw forecasts, there are remaining biases of up to 0.7 K for the lead-0 and lead-7 forecasts.

The effect of the BCSD on the root mean squared errors, which are shown in Fig. 3 is much more diverse. In general, the RMSE of SEAS5 BCSD is lower compared to the raw forecasts. As an example, the RMSE of the precipitation and radiation forecasts for February over the CC-basin are reduced from around 6 mm/day and 30 W/m$^2$ and to less than 4 mm/day and 20 W/m$^2$, respectively. The minimum temperature RMSE during the rainy season of the KA-basin could be reduced from 3 K to less than 2 K.

However, there are other cases where the bias-correction shows almost no improvement. While the biases of the raw precipitation, temperature, or radiation forecasts for the SF-Basin are much lower after applying the BCSD, the RMSE remains

almost unchanged. The same holds true for the precipitation forecasts for the BN-Basin. Moreover, there are still some lead-dependent effects of the bias-corrected forecasts, which can be seen from the gaps and jumps in the RMSEs of the temperature and radiation forecasts over the SF-Basin.

## 4.2 Topographic and resolution-dependent effects

To evaluate the impact of the improved spatial resolution, Fig. 4 shows the total accumulated precipitation and its standard deviation during the four months of the rainy season from the raw and bias-corrected lead-0 forecasts, respectively, and ERA5-Land. The precipitation sums of the raw forecasts are generally lower than those from ERA5-Land. This is particularly visible for the mountainous headwaters of the KA (D01) and CC (D04) basins, where the raw forecasts predict seasonal precipitation sums of less than 500 mm (D01) and 2000 mm (D04), respectively, while ERA5-Land show values of more than 750 mm (D01) and 2400 mm (D04), respectively. Furthermore, there is a single connected precipitation pattern along the Andes in the D04-domain and multiple precipitation bands in the D03-domain over the Ethiopian highlands. ERA5-Land, however, shows much more distinct spatial precipitation peaks, which agree very well with the complex topography in the mountainous headwaters. After applying the BCSD, the patterns and values of the seasonal forecasts match almost perfectly with the reference data and, hence, also the higher resolved topography. The same holds true for the standard deviation of seasonal precipitation. The raw forecasts tend to underestimate the precipitation variability across all four domains. Especially in the mountainous areas of D01 and D04, ERA5-Land and, hence, SEAS5 BCSD show maximum standard deviations of $\pm 600$ mm and more while the raw forecasts only reach values of less than $\pm 450$ mm. On the other hand, the raw forecasts show a higher precipitation variability particularly across the South-Western corner of D03, which is reduced in SEAS5 BCSD and therefore agrees better with ERA5-Land.

## 4.3 Lead-time dependent effects

The BCSD-approach further corrects for lead-dependent effects. The magnitude of these effects strongly depends on the lead time, which can be seen when comparing, e.g., the climatologies of the raw forecasts from different issue months. As an example, Fig. 5 shows the difference between the July-forecasts for D03 from different issue months. These differences obviously increase with increasing lead time. While precipitation amounts are decreasing for higher lead-times, temperatures and radiation are increasing.

One reason for these drifts is a shift of higher temperatures and higher radiations with increasing lead times towards south. This is visible in Fig. 6, which shows the July-forecasts from SEAS5 and SEAS5 BCSD from different lead times, compared to ERA5-Land. Despite the biases in absolute magnitudes, the climatology of the higher-lead SEAS5 temperature and radiation forecasts match better with the climatology from ERA5-Land. The differences between the lead-0- and lead-5-forecasts in Fig. 5 also show the largest deviations in the Northern part of D03.

In most other cases, however, the climatologies from lower lead times show a better agreement with the ERA5-Land climatology.

The SEAS5 BCSD forecasts show only minor lead-dependent effects (Fig. 5). The remaining differences for precipitation, temperature and radiation between the low- and higher-lead forecasts are below 0.5 mm/day, 0.5 K and 10 W/m$^2$, respectively. Similarly, as depicted in Fig. 5, the lead-0 and lead-5 forecasts for precipitation, temperature, and radiation forecasts in July as well as the ERA5-Land based estimates agree almost perfectly in magnitude and spatial patterns, indicating that the model drift of SEAS5 is almost completely removed after applying the BCSD-approach. This is also true for the other three study domains (not shown).

## 4.4   Wet and dry day frequencies

Besides biases in the absolute values from raw forecasts, we usually also have to take into account biases in the frequencies of wet and dry days. Figure 7 shows the wet day probability from the lead-0- and lead-5-forecasts, respectively, for a single month for all four domains. Similar to the drifts in the absolute values and patterns, there is also a clear difference between the wet-day-probabilities from different lead-times. For example, the lead-0-forecasts for June over the D03-domain predict a wet-day probability during June of about 100 % across large parts of the Ethiopian highlands. This means that there is at least 1 mm of precipitation on every single day in June. In the lead-5-forecasts (which are issued in January), this probability is reduced to 80 % and less. In other words, only 80 % of the forecasted days in June receive precipitation amounts of at least 1 mm per day. Similarly, the higher-lead-forecasts over the D01-domain also predict lower wet-day-probabilities across the Zagros mountains. After correcting for this lead-dependent wet day frequency, the BCSD forecasts show spatial patterns very similar to the reference data and more consistent frequencies across the different lead-times.

## 4.5   Overall performance

The change in overall performance of the seasonal forecasts due to the bias-correction and spatial disaggregation with respect to ERA5-Land is evaluated with the *Continuous Ranked Probability Skill Score* (CRPSS B). In general, the overall performance of SEAS5-BCSD improves compared to raw SEAS5, i. e., the cumulative distribution functions (CDFs) of SEAS5-BCSD better correspond to the reference ERA5-Land than the CDFs of raw SEAS5 (Fig. 8). Largest improvements for all basins are found for the minimum temperature, with frequent CRPSS values $> 0.4$ indicating an improvement in the distributional distances by 40% compared to raw SEAS5. Among the basins, largest improvements by BCSD are produced for the CC-Basin, especially for precipitation and maximum temperature. For the TA and BN basins, the above mentioned lead dependent effect is evident with larger improvements for the lower lead-times of the temperature and radiation forecasts. For the KA-Basin, precipitation forecasts for November and April may be worsened in their performance by BCSD. In contrast, the main four months of the rainy season of KA show improvements mainly above 30%. Also for the SF basin, the December precipitation forecasts may be worsened, whereas other months only show little improvement of SEAS5 BCSD compared to raw SEAS5 for precipitation forecasts. Similarly, there is only slight improvement for the maximum temperature forecasts for the KA-basin or the higher-lead temperature forecasts from December to March for the CC-basins.

## 5   Discussion

The BCSD forecasts show a much better agreement with ERA5-Land as the raw SEAS5 product across most variables, domains and forecast months. However, to understand the performance of the BCSD across the different study regions, the regional climatic conditions are important.

Over regions like the East-African D03-domain, the rainy season is dominated by the East African monsoon. This is usually associated with daily convective precipitation and, hence, many continuous wet days during the rainy season. According to Fig. 6 and 7, the African monsoon is predicted with higher rainfall intensities, higher wet day frequencies, and especially towards the northern parts of the domain with lower temperature and radiation at shorter forecast horizons than at longer lead-times. Hence, it is assumed that in contrast to the model climatologies from SEAS5 and ERA5-Land, the initial conditions (which strongly influence the low-lead forecasts) cause the low-lead forecast to show higher intensities as well as a more Northern extension of the summer monsoon. The comparison with the reference ERA5-Land reveals that the spatial extent of the monsoon is predicted too far towards the North at low lead-times. However, the rainfall intensity and wet day frequency is more realistic than at long forecast horizons.

Such spatial and temporal inconsistencies in the forecasted spatial extent and intensity of the monsoon from different issue dates impede the direct application of raw forecasts for the regional water management. Therefore, as we correct the raw forecasts to the same ERA5-Land reference data across all lead times, this lead-dependency is eliminated during the bias-correction. This simplifies the use as well as the interpretation of our BCSD forecasts compared to the raw SEAS5 products.

For some regions and variables, e.g., for the maximum temperature forecasts across the KA-basin, the precipitation or radiation forecasts across the SF-basin or the precipitation forecasts across the BN-basin, almost no or little improvements for most forecast months can be identified, indicated by CRPSS-values of around 0 in Fig. 8 and almost identical RMSE-values of SEAS5 and SEAS5 BCSD in Fig. 3. In such cases, there is only a limited effect of the bias-correction which can be explained by, e.g., an already good correspondence between the raw forecasts and ERA5 Land (indicated by low biases in Fig. 2) and/or rather random biases. According to, e.g., Thrasher et al. (2012), variables that are systematically biased usually benefit more from a quantile-mapping based bias correction than randomly biased variables.

Hence, large improvements of overall performance, indicated by high CRPSS-values (Fig. 8), usually point to large systematic discrepancies between the raw SEAS5 and the reference ERA5-Land. This is obvious for, e.g., the minimum temperature forecasts which show a negative bias of the raw forecasts (Fig. 2), high CRPSS values (Fig. 8) and reduced RMSEs after the bias-correction (Fig. 3) across almost all basins and forecast months.

Besides these mostly positive results for BCSD, the mixed impact of the BCSD-approach across the D01-domain, indicated by negative CRPSS-values for the KA-basin in Fig. 8, requires a closer look. Iran's climate during the rainy season is dominated by migrating low-pressure systems mainly from the West and the Mediterranean sea (Khalili and Rahimi, 2014). The precipitation over the D01-domain hence occurs intermittently and spatially variable, which is usually difficult to predict especially with higher lead-times and over the mountainous headwaters of the Karun. For such regions, it is necessary to correct for the amount and spatial location of precipitation as well as the wet and dry day frequency. While, according to Fig. 7, the wet day

frequency even for high lead-times could be improved, the CRPSS values in Fig. 8 show worse agreement with ERA5-Land after the bias-correction during the transition from the dry to the wet season (November) or the transition from the wet to the dry season (April). We assume that this is caused by the application of a 31-day-window for estimating the distribution functions, which might not be adequate in such strongly varying climate conditions during the transition months between the wet and dry seasons. This can also be seen in Fig. 2, where particularly the lead-0 and lead-7 temperature forecasts show remaining biases with values of up to 0.7 °K. The temperature bias in the first and last forecast months appears strongest in the KA-Basin due to the large annual temperature variations with an annual temperature range of up to 45° (Table 1). Nevertheless, mostly positive CRPSS values at these lead-times (Fig. 8) result from still reduced biases compared to the raw forecasts (Fig. 2).

In general, during the first and last days of a forecast, we can not fill the complete 31-day-window for estimating the forecast CDF. As an example, the reference CDF for the first of January is based on the values from the 17th of December till the 16th of January, while the CDF of the January forecast for the 1st of January only uses the values from the 1st to the 16th of January. If there are strong temporal climate gradients or heteroscedasticity, e.g., during the transition from a cold to a warm period, a bias-correction using moving windows can lead to remaining biases and, hence, to statistical inconsistencies particularly on non-daily timescales. An approach to account for such gradients would be to use a dynamic moving window, where the length of the window is based on, e.g., the gradient of the daily climatology. It will be subject of future studies if such an approach is able to improve the statistical consistency particularly during the first and last days of the forecast.

The representation of small scale features in SEAS5 BCSD, particularly in complex and mountainous terrain, benefits from the explicit altitude correction in ERA5-Land, which was necessary due to the higher spatial resolution compared to ERA5: when correcting the SEAS5 forecasts towards such a reference, we automatically include an indirect correction for altitude. For the small basins of KA and CC with large altitude differences, the added value of spatial disaggregation and bias correction (indirect altitude correction and better representation of small scale features) is therefore most evident. Particularly at high elevations of the Zagros Mountains in KA and the Andes in CC, the higher resolution and subsequent bias-correction allows for locally distinct precipitation intensities (Fig. 4). Also in the Ethiopian Highlands of D03 higher resolution produces more complex (at this resolution circular shaped) structures around the Ethiopian mountains. Independent of the accuracy of the seasonal forecasts, we strongly assume that the higher spatial resolution and, hence, better representation of small-scale precipitation patterns make the BCSD SEAS5 forecasts more suitable for the regional water management. As already shown in, e.g., Westrick and Mass (2001), a higher spatial resolution of the atmospheric forcing (i.e., precipitation) usually leads to more accurate streamflow modeling.

We would also like to discuss the choice of the bias correction method used. As reported by, e.g., Anghileri et al. (2019), bias-correction is crucial to improve both forecast quality and value. The quantile mapping method that is used in this study serves this purpose and is a widely used, well understood and robust technique that is not computational demanding and can be easily implemented (Siegmund et al., 2015). During the recent years, there have been numerous studies in which new approaches were presented. While other techniques can lead to more skillful, reliable and accurate forecasts (e.g., Schepen et al., 2018; Manzanas et al., 2019; Khajehei et al., 2018) or lower biases (e.g., Alidoost et al., 2019) as quantile mapping for example tends to produce negatively skillful forecasts when the raw forecasts are not significantly positively correlated with observations

(Zhao et al., 2017), it should be considered that quantile mapping still serves as the reference method in most of the recent bias correction studies. In other words, there is currently no other bias correction method that is similarly widespread. This not only improves the comparability of our data with similar studies, but also marks our SEAS5 BCSD forecasts as a reference product for exploring new forecast products and developing and evaluating new bias correction techniques.

## 6   Conclusions

In this study we present a comprehensive dataset of bias-corrected and spatially disaggregated seasonal forecasts for four different semi-arid domains across three continents. The forecasts are based on the most recent version of ECMWF's seasonal forecast system SEAS5, which are corrected towards the ERA5-Land land-surface re-run of the ERA5-reanalysis with an enhanced spatial resolution of $9\,\mathrm{km}$ (here: $0.1°$). The final SEAS5 BCSD repository contains seasonal forecasts at daily and monthly resolution for precipitation, mean, minimum and maximum temperature as well as for radiation for the period from 1981 till 2019 with a spatial resolution of $0.1°$. For each of the 468 issue dates, we provide ensemble forecasts with 25 or 51 members (since 2017) for the coming 214 days. Currently, the data covers domains in Iran (D01), Brazil (D02), Ethiopia/Sudan (D03) and Ecuador/Peru (D04), but it is planned to extend this list to further domains.

The comparison of our SEAS5 BCSD product with the raw SEAS5 forecasts against reference data from ERA5-Land clearly indicated a reduction of biases and root mean squared errors across most study regions and variables. Further, the spatial resolution of the forecasts is improved from $36\,\mathrm{km}$ to $0.1°$ and the patterns of precipitation, temperature and radiation show much better agreement with the reference data. Finally, model drifts are reduced which leads to temporally more consistent forecasts.

Besides these improvements, we could also observe remaining biases after bias-correction particularly during the low- and high-lead temperature forecasts for the KA-basin. This is explained with a highly variable climate with strong gradients and heteroscedasticity, where a moving window can introduce statistical inconsistencies when, e.g., monthly averages are derived from daily data. The impact of a highly dynamic climate on the statistical consistency of bias-corrected forecasts obviously has also huge implications to, e.g., global approaches. Therefore, future works have to further examine methods and approaches to account for such strong gradients in the reference climatologies.

Our bias-corrected and spatially disaggregated seasonal forecasts are freely available on both daily and monthly temporal resolution via the World Data Center for Climate. To our knowledge, this is the first multi-variable and multi-domain high-resolution seasonal forecast for a period of almost 40 years. It provides an unique product for a wide variety of researchers, stakeholders and other experts from the water sector who are interested in a consistent dataset of post-processed seasonal forecasts. This product gives local experts from the four study domains, who often do not have the computational framework conditions or access to the operational products from ECMWF, the opportunity to investigate the potential of high-resolution seasonal forecasts for, e.g., the regional water management, drought forecasting or irrigation planning. Derived products like categorical forecasts, based on our SEAS5 BCSD product, are therefore already adopted by several weather services and other higher-level authorities in the study domains. As SEAS5 BCSD also covers all months during the re-forecasting period

since 1981, it can be used to review and refine currently existing decision calendars and dates when actions and management strategies for the coming rainy season are defined.

That being said, the published SEAS5 BCSD dataset currently contains only the daily and monthly ensemble forecasts and no derived information like, e.g., probabilistic and categorical forecasts or drought indicators. Hence, for transferring this product into practice, we have to a) identify and compute regionally suitable forecasting measures and indicators, b) make this information accessible, e.g., via an user-friendly online platform and c) ensure a proper communication and dissemination of the potential but also the limitations of such current seasonal forecasting products to all end-user sectors like authorities, water managers or farmers.

For these subsequent steps, SEAS5 BCSD can be used as consistent data resource and, hence, serves as a contribution towards ensuring a sustainable and timely regional water management in our semi-arid study regions. Ultimately, in the context of global climate change with increasing risks of climatic extreme events, we need to ensure that longer-term forecasts like SEAS5 BCSD are incorporated by authorities and included in the water resources planning for improving future disaster preparedness.

## 7 Data availability

The bias-corrected and spatially disaggregated seasonal forecasts are published via the World Data Center for Climate (WDCC), which is hosted by the the German Climate Computing Center (DKRZ), within the project *Seasonal Water Resources Management for Semiarid Areas: Regionalized Global Data and Transfer to Practise* (SaWaM, https://cera-www.dkrz.de/WDCC/ui/cerasearch/project?acronym=SaWaM). In this project, we have created the four experiments SaWaM D01, SaWaM D02, SaWaM D03, and SaWaM D04 (i.e. one experiment for each study domain), which contain all products for the respective region. Our SEAS5 BCSD forecasts are available via the dataset group SaWaM SEAS5 BCSD, which contains all daily and monthly forecasts:

- SaWaM D01 SEAS5 BCSD (https://doi.org/10.26050/WDCC/SaWaM_D01_SEAS5_BCSD): Seasonal Water Resources Management for Semiarid Areas: Bias-corrected and spatially disaggregated seasonal forecasts for the Karun Basin (Iran) (Lorenz et al., 2020b)

- SaWaM D02 SEAS5 BCSD (https://doi.org/10.26050/WDCC/SaWaM_D02_SEAS5_BCSD): Seasonal Water Resources Management for Semiarid Areas: Bias-corrected and spatially disaggregated seasonal forecasts for the Rio São Francisco Basin (Brazil) (Lorenz et al., 2020c)

- SaWaM D03 SEAS5 BCSD (https://doi.org/10.26050/WDCC/SaWaM_D03_SEAS5_BCSD): Seasonal Water Resources Management for Semiarid Areas: Bias-corrected and spatially disaggregated seasonal forecasts for the Tekeze-Atbara and Blue Nile Basins (Ethiopia/Eritrea/Sudan) (Lorenz et al., 2020d)

– SaWaM D04 SEAS5 BCSD (https://doi.org/10.26050/WDCC/SaWaM_D04_SEAS5_BCSD): Seasonal Water Resources Management for Semiarid Areas: Bias-corrected and spatially disaggregated seasonal forecasts for the Catamayo-Chira Basin (Ecuador/Peru) (Lorenz et al., 2020a)

Each of these four groups contains six datasets: BCSD_daily_pr and BCSD_monthly_pr (daily and monthly precipitation forecasts), BCSD_daily_tas and BCSD_monthly_tas (daily and monthly average, minimum and maximum temperature forecasts), BCSD_daily_rsds and BCSD_monthly_rsds (daily and monthly surface solar radiation forecasts). All datasets contain forecasts from the issue date (i.e. the first of each month) for the next 215 days (daily) and 6 months (monthly), respectively.

Users interested in a *teaser product* are advised to use the monthly averaged forecasts. They have a maximum download size of around 2 GB for precipitation, 2.4 GB for radiation and 6 GB for the three temperature variables over our largest domain across Brazil for the whole period from 1981 to 2019 and all ensemble members. The data size for the other domains is of course much smaller. Some of the products as well as derived forecast measures like categorical precipitation and temperature forecasts are visualized through an online decision support system for the regional water management at https://sawam.gaf.de/. This system is currently under joint development with the company GAF AG (https://www.gaf.de, Munich, Germany). As of now, forecasts for the Brazilian and Sudanese and Ethiopian domain are included. The data for Iran will be implemented in the near future.

We also publish the BCSD forecasts through the Karlsruhe Institute of Technology (KIT) - Campus Alpin THREDDS Server. In contrast to the products available via the WDCC, the operational forecasts are only available via the THREDDS-Server. These are published with a delay of about 1 day after the release of the official seasonal forecasts from ECMWF on the fifth of each month. For getting access to the operational products, contact Christof.Lorenz@kit.edu.

## Appendix A: Empirical quantile mapping

### A1  Empirical quantile mapping

We follow the classical empirical quantile mapping approach as depicted in, e.g., Wood et al. (2002); Voisin et al. (2010). The bias-correction is performed separately for each single pixel. For each forecasted day, we select a window of $\pm$ 15 days around the forecasted day from all ensemble members and all re-forecasts of the period 1981 to 2016 which have been initialized in the same month. The CDF for the forecasts is then computed from this large sample. As an example, the CDF for the 1st of January from the January forecast is based on 16 (days in January) $\times$ 25 (ensemble members) $\times$ 36 (years) = 14400 values. The CDF for the 31st of January from the same issue month is based on 27900 values (as we use the 15 days before and after the 31st of January). Obviously, during the first and last 15 days of a forecast, we have to use less samples as we cannot extend the window before or after the forecasted period. While this might introduce inconsistencies particularly during the first and last days of a forecast, we think that this approach of a daily moving window is still more climatically reasonable than estimating a single CDF for each month. This approach, while computationally less expensive, can lead to large jumps between consecutive days at the end of the previous and beginning of the next month. The CDF for the reference data is estimated in the same way

by using 31 (days) $\times$ 36 (years) = 1116 samples for each forecasted day. Using the CDFs for the forecasts $F_{i,\theta,\text{mod}}$ and the reference $F_{i,\theta,\text{ref}}$ for the forecasted day $i$ and pixel $\theta$, the bias-correction of each forecasted value $X_{i,\theta}$ is performed through

$$\widehat{X}_{i,m,\theta} = \quad F_{\text{doy,ref}}^{-1}(F_{\text{doy,mod}}(X_{i,m,\theta})) \tag{A1}$$

This is shown exemplarily in Fig. A1a.

## A2 Bias-correction of extremes

As we are using ensemble-based forecasts, we usually have more samples from which we can estimate the forecast CDF compared to the reference data. If further an empirical quantile mapping is applied, we can get forecasted probabilities outside the range of the empirical reference quantiles (i.e., below $1/(n+1)$ or above $n/(n+1)$, where $n$ is the number of samples from which the reference CDF is derived; upper dashed line in Fig. A1). To apply the bias-correction to such extreme values, some extrapolation is required. Here, we use the *constant correction* method from Boé et al. (2007). We first calculate the correction that corresponds to the lowest and highest reference quantile, respectively. These constant corrections are then applied to all forecast values below or above the lowest and highest reference quantile. Due to its simple application and robustness, this approach is a good choice particularly in cases where no climate change signal is expected or when the parametric distributions for the variables to be corrected are unknown or difficult to estimate.

## A3 Correction of the precipitation intermittency

Besides the correction of the *absolute* values, there might also be a bias between the wet- and dry-day frequencies. For correcting these frequencies, we apply the same approach as in Voisin et al. (2010). We first compute the dry-day probability of the forecasts and the reference data. If a forecast falls below the reference dry-day probability, it is set to zero. This is exemplary demonstrated in Fig. A1c. If, however, the dry-day probability of the forecasts is higher than the reference data and we obtain a zero-precipitation forecast, we draw a uniform random sample between 0 and the dry-day probability of the forecasts (depicted by the arrow in Fig. A1d). If this value is below the dry-day probability of the reference (the lower dashed line in Fig. A1d), the forecast is again set to zero. If not, it is corrected using the inverse CDF of the reference.

## A4 Consistent correction of minimum, maximum, and average temperature

Due to the very nature of QM-based bias-correction, the corrected temperatures can become physically unrealistic as

1. the corrected minimum (maximum) temperatures are higher (lower) than the corrected maximum (minimum) temperatures or

2. the corrected average temperature is higher (lower) than the corrected maximum (minimum) temperature, respectively.

The first two cases are comprehensively discussed in Thrasher et al. (2012). For bias-correcting daily temperature data, they propose to first calculate the diurnal temperature range from the maximum and minimum temperature. The bias-correction is

then applied to the maximum temperature and diurnal temperature range (instead of the minimum temperature). Afterwards, the corrected minimum temperature is derived by subtracting the bias-corrected diurnal temperature range from the bias-corrected maximum temperature. As we are interested in correcting the average temperature as well, we apply a slightly modified version of this approach. Instead of computing the diurnal temperature range, we compute the difference between the maximum (minimum) and average temperature, respectively. Then, we apply the bias-correction to the two ranges as well as the average temperature. Corrected maximum and minimum temperatures are then computed by adding and subtracting the ranges to and from the corrected average temperature, respectively.

## Appendix B: Verification metrics

To compare the bias-corrected and spatially disaggregated forecasts against the reference data, we use three different verification metrics. First, for evaluating the overall level of agreement of the corrected data, we compare the BCSD and raw SEAS5 forecasts against the ERA5 Land dataset using the bias and Root Mean Squared Error (RMSE). For a pixel or basin average, these are

$$Bias \quad = \frac{1}{N}\left(X_p - X_o\right) \tag{B1}$$

and

$$RMSE \quad = \sqrt{\frac{1}{N}\left(X_p - X_o\right)^2}, \tag{B2}$$

respectively, where $X_p$ and $X_o$ are the predictions and reference values, respectively and $N$ the number of samples. In our case, for monthly forecasts, $N$ is 36 (years). Obviously, the Bias and RMSE can be computed for different lead-times, which then gives information about the dependency of the error from the lead-time.

While the Bias and RMSE is suitable for comparing the overall agreement of the ensemble mean against a reference, there is a wide range of metrics particularly for ensemble forecasts. A comprehensive overview and discussion of ensemble forecast verification measures can be found in, e.g., Casati et al. (2008). In this study, we use the Continuous Ranked Probability Skill Score (CRPSS, e.g Hersbach, 2000) for the evaluation of overall performance. The CRPSS is the continuous extension of the widely used Brier Skill Score (BSS Brier, 1950) and compares relative distributional distances of forecast and reference data. While the BSS aims at the verification of specific events like the probability of precipitation amounts $> 10\,\mathrm{mm/day}$, the CRPSS extends this to *all* possible events. As a skillscore, comparing the prediction skill of different forecasts, the CRPSS is defined as

$$CRPSS \quad = 1 - \frac{CRPS_{\text{forecast}}}{CRPS_{\text{reference forecast}}}. \tag{B3}$$

Here, the Continuous Ranked Probability Score ($CRPS$) for a forecasted quantity $x$ is defined as

$$CRPS(F_p, x) \quad = \int_{-\infty}^{+\infty} \left(F_p\left(x\right) - F_o\left(x\right)\right)^2 dx \tag{B4}$$

where $F_p(x_p)$ and $F_o(x_o)$ are the cumulative distribution functions from the ensemble seasonal forecasts and the reference data, respectively. In this study, we use the empirical CDF for approximating the ensemble CDF of the forecasts and the CDF of the reference data is defined as

$$F_o(x) \quad = H(x - x_o) \tag{B5}$$

where $H(x)$ is the Heaviside-function. Obviously, for each observation, we compute a single $CRPSS$. For evaluating the $CRPSS$ across multiple points in time and/or space, we calculate the average from the single $CRPSS$ values.

In most cases, the comparison with the reference forecast in the denominator of the $CRPSS$ simply uses climatology. However, in this study, we are interested in the increase in the level of agreement of the SEAS5-forecasts with the ERA5-Land reference data after applying the BCSD. We therefore compute the skill score from the $CRPS$ between the SEAS5 BCSD and the raw SEAS5 forecasts against the ERA5-Land reference, respectively.

*Author contributions.* C.L. collected and processed the seasonal forecasts from ECMWF, developed and applied the bias-correction and spatial disaggregation algorithm, computed the final seasonal forecast products, initiated the data dissemination via DKRZ WDCC and the KIT Campus Alpin THREDDS Server and implemented the operationalization of the presented approach; C.L. and T.P. conducted the evaluation of the BCSD forecasts and wrote the first draft of the paper; C.L., T.P., P.L. and H.K. reviewed the paper and prepared the final version of the manuscript.

*Competing interests.* The authors declare that they have no conflict of interest.

*Disclaimer.* TEXT

*Acknowledgements.* The authors would like to thank the German Federal Ministry of Education and Research (BMBF) for funding this research within the Seasonal Water Resources Management: Regionalized Global Data and Transfer to Practise (SaWaM, http://grow-sawam. org) project. We would also like to thank the ECMWF and the Copernicus Climate Service for providing the SEAS5 seasonal forecasts as well as the IT-team of the Karlsruhe Institute of Technology - Institute of Meteorology and Climate Research for providing access to the HPC-environment. Finally, we would like to thank the DRKZ and WDCC teams and, particularly, Mr. Heinrich Widmann and Dr. Andrea Lammert for uploading, publishing, and supporting our product.

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

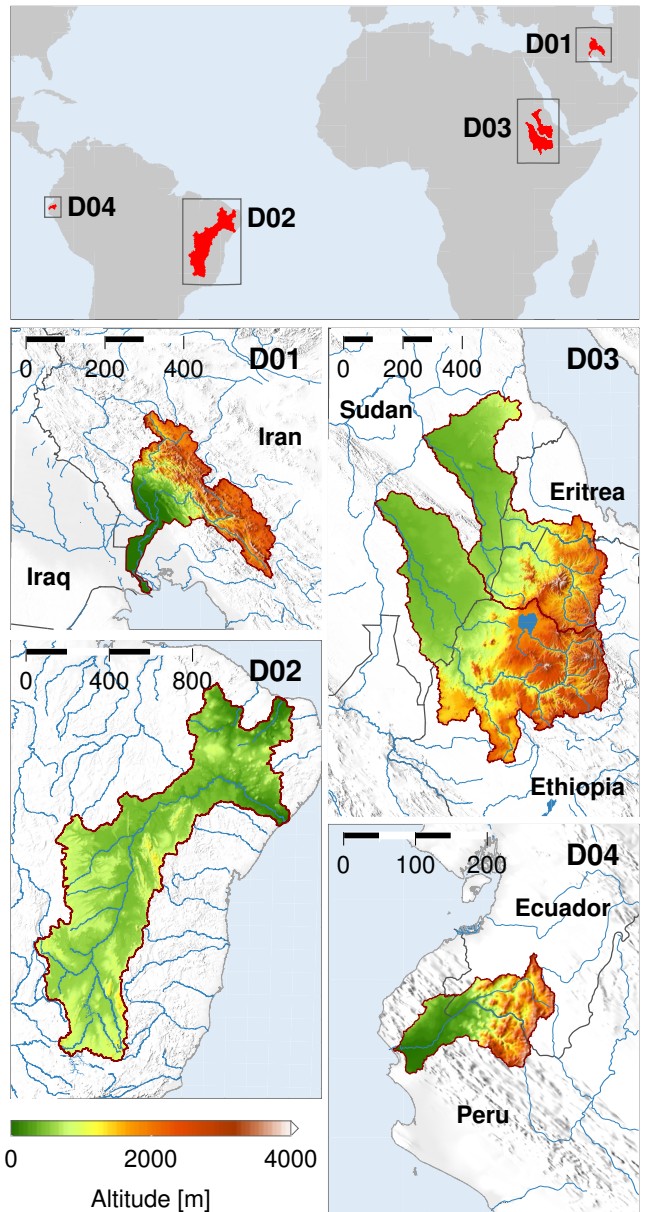

**Figure 1.** Overview of the four study domains: the Karun Basin (KA, Iran, D01, top left), the Extended São Francisco Basin (SF, Brazil, D02, bottom left), the Tekeze-Atbara (to the North) and Blue Nile (to the South) Basins (TA and BN, Sudan, Ethiopia and Eritrea, D03, top right), and the Catamayo-Chira Basin (CC, Ecuador and Peru, D04, bottom right). The distance scales in the top left corner of the maps are given in units of kilometers. The basin topography is based on the high-resolution ETOPO1 Global Relief Model (Amante and Eakins, 2009) while coastlines, rivers and political borders are taken from the *Global Self-consistent, Hierarchical, High-resolution Geography Database* (GSHHG, Wessel and Smith, 1996). The basin boundaries are based on the HydroSHEDS dataset (Lehner and Grill, 2013) with some slight modifications and adjustments for ensuring the consistency with boundary definitions from local authorities and stakeholders from the study regions.

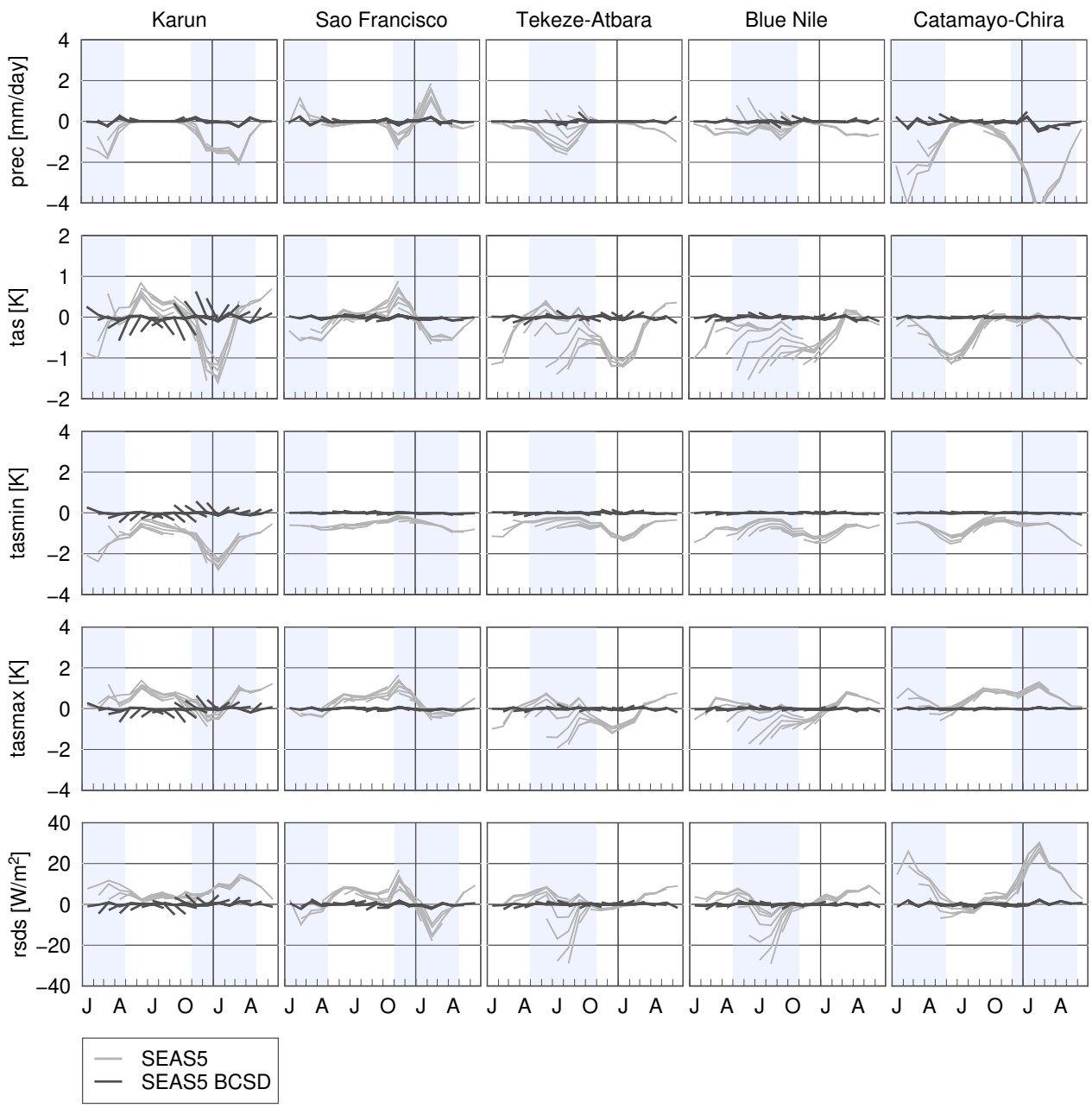

**Figure 2.** Bias of monthly (from top to bttom) precipitation (pr), average temperature (tas), minimum temperature (tasmin), maximum temperature (tasmax) and radiation (rsds) forecasts from SEAS5 (grey) and SEAS5 BCSD (black) with respect to ERA5 Land during the period 1981 to 2016 for all five study basins from 12 issue months (January to December). The vertical line between December and January indicates the end of the year; the values right of this line belong to the higher lead forecasts issued in September till December. The blue-shaded area depicts the six months of the rainy season.

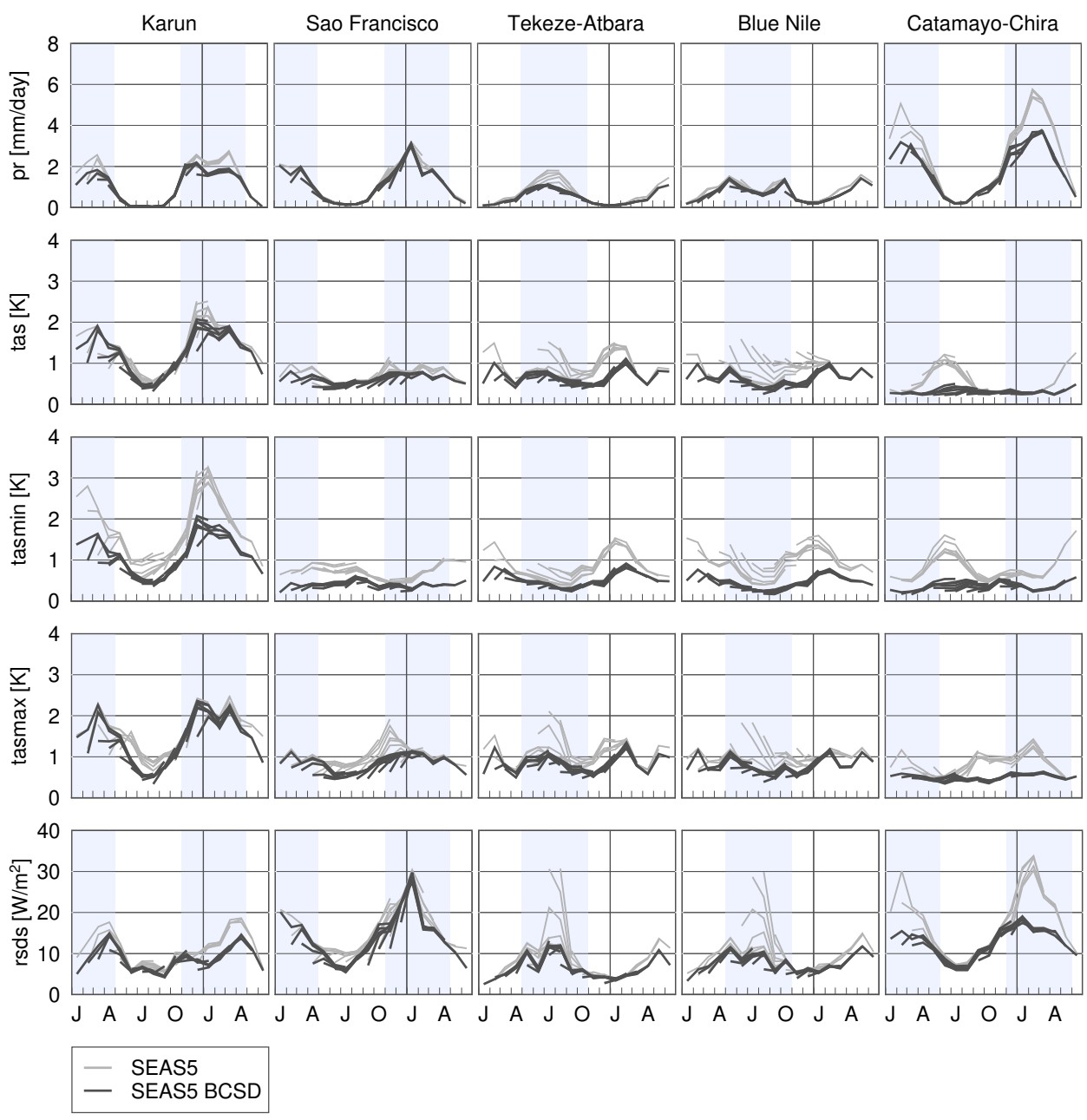

**Figure 3.** Same as Fig. 2 but for the RMSE of SEAS5 and SEAS5 BCSD compared to ERA5-Land.

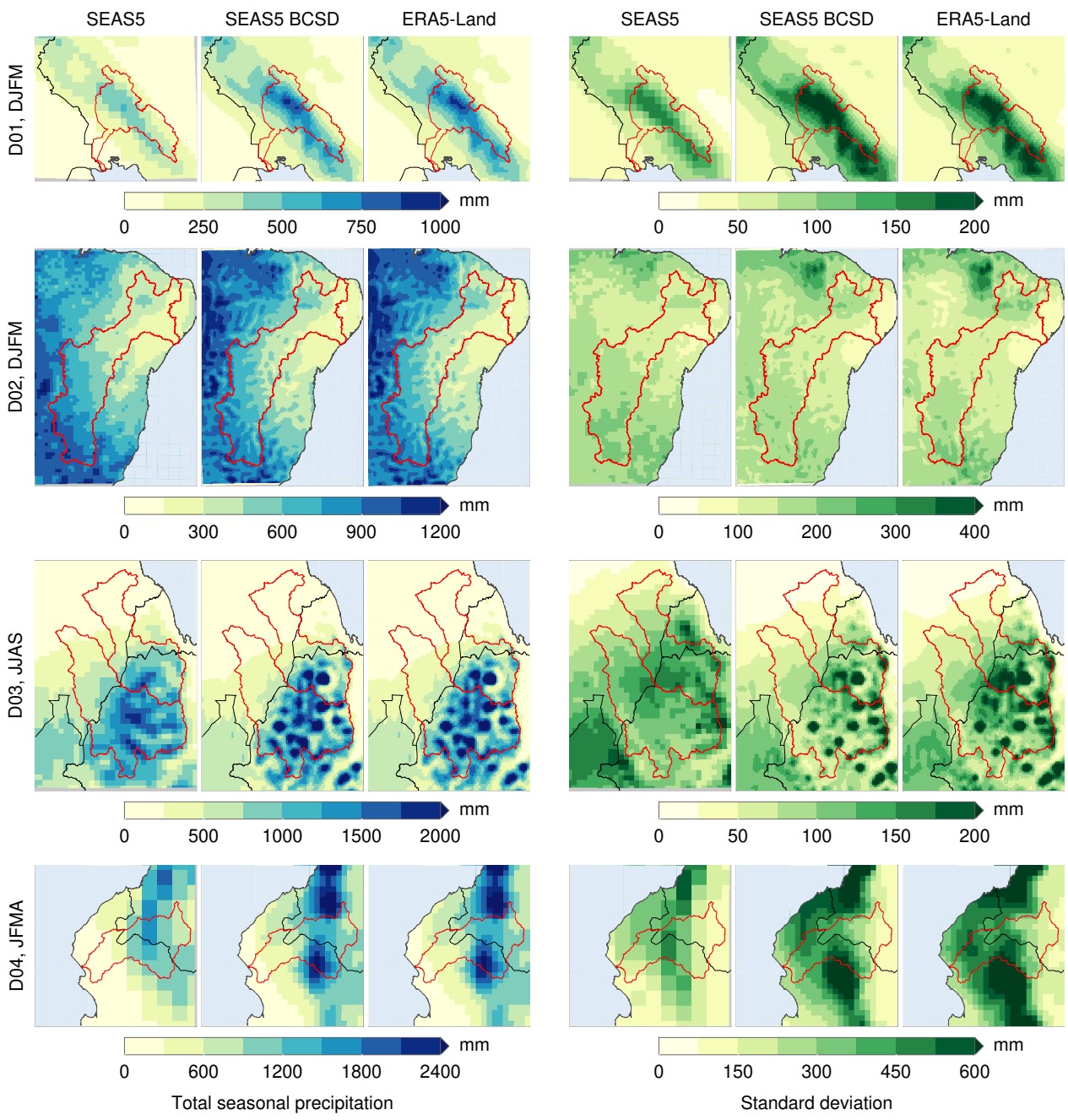

**Figure 4.** Total precipitation (left three columns) and the corresponding standard deviation (right three columns) during the four main months of the rainy season for the four study domains from raw SEAS5 and SEAS5 BCSD lead-0 forecasts and ERA5-Land, averaged over the period 1981 to 2016.

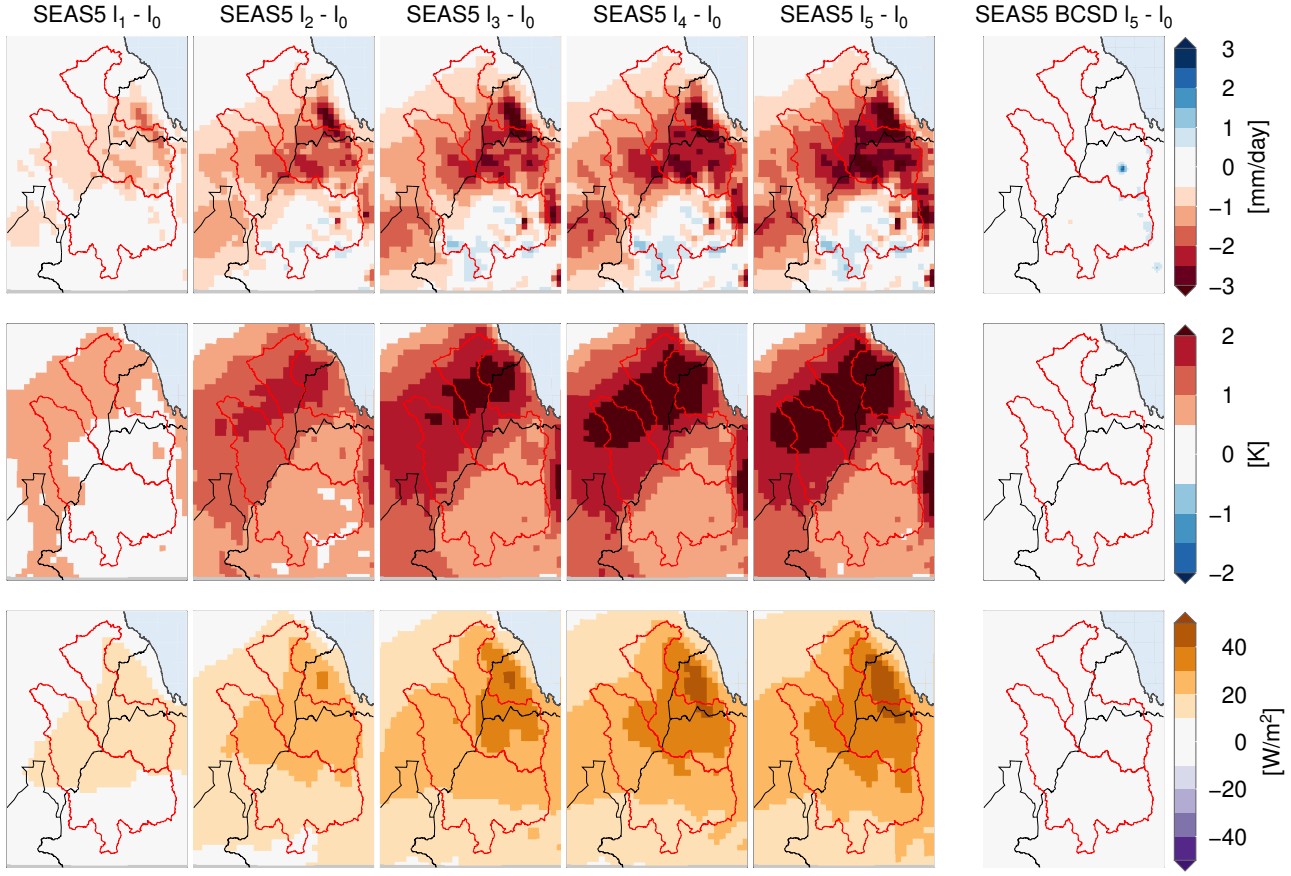

**Figure 5.** Differences between the July-forecasts from different issue months for precipitation (top row), temperature (middle row) and radiation (bottom row) for the African domain D03.

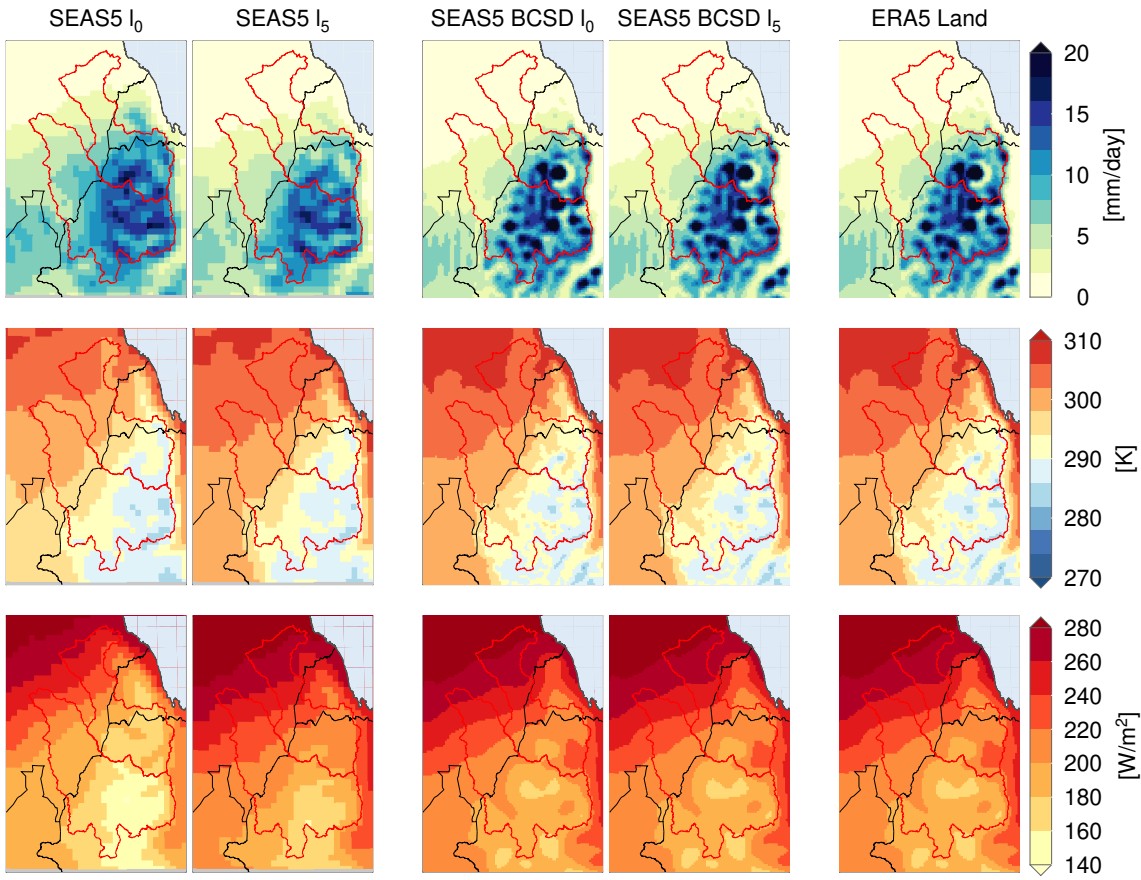

**Figure 6.** Average July predictions from SEAS5 lead 0 and 5 (first two columns), SEAS5 BCSD lead 0 and 5 (center columns) and reference values from ERA5-Land (right column) for precipitation (top row), average temperature (middle row) and radiation (bottom row) for the African domain D03.

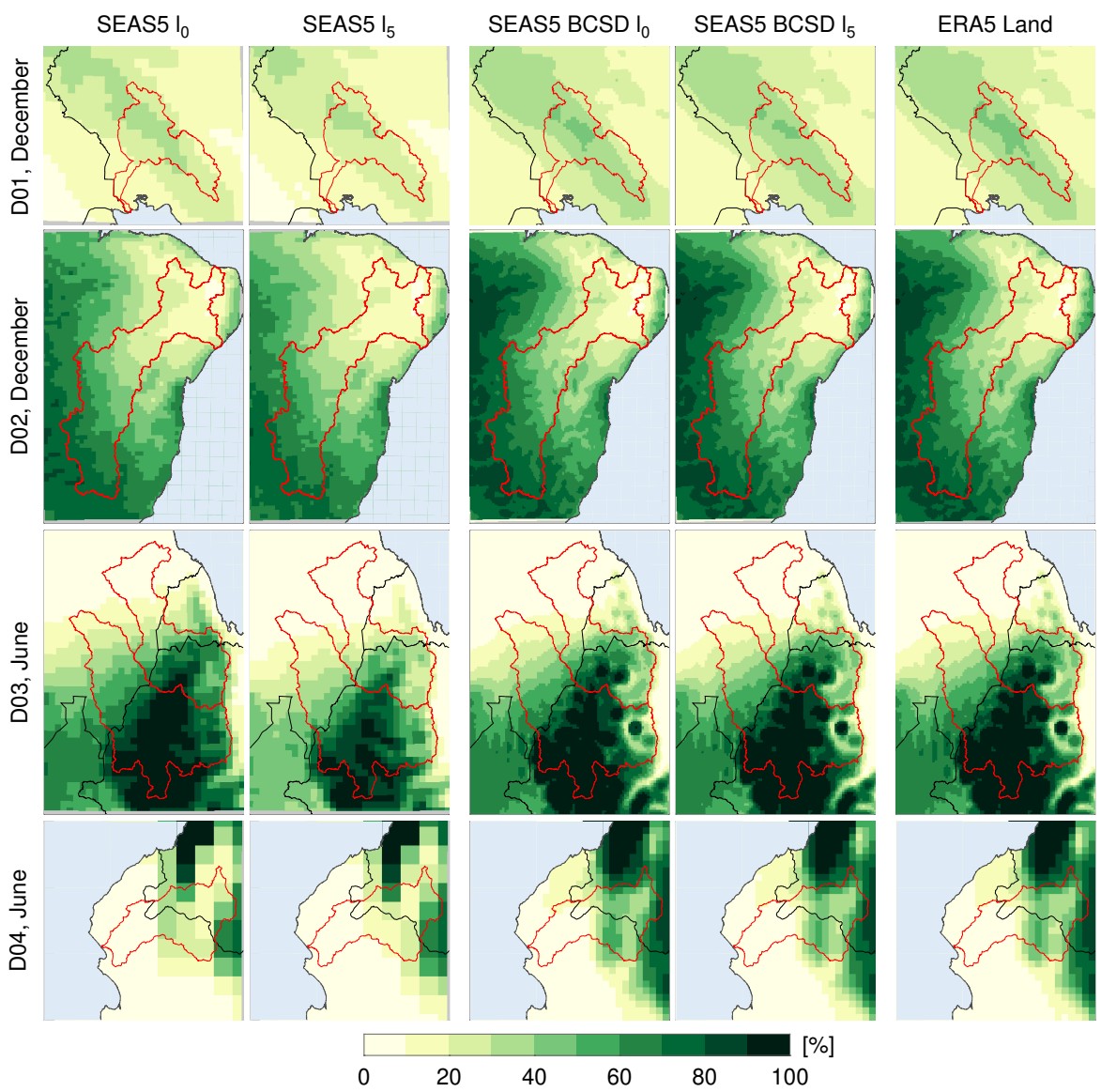

**Figure 7.** Wet day ($> 1$ mm/day) probability for the four study domains for a single month (D01: December, D02: December, D03: June, D04: May) from SEAS5, SEAS5 BCSD for both lead-0 and lead-5, respectively, and the reference ERA5-Land.

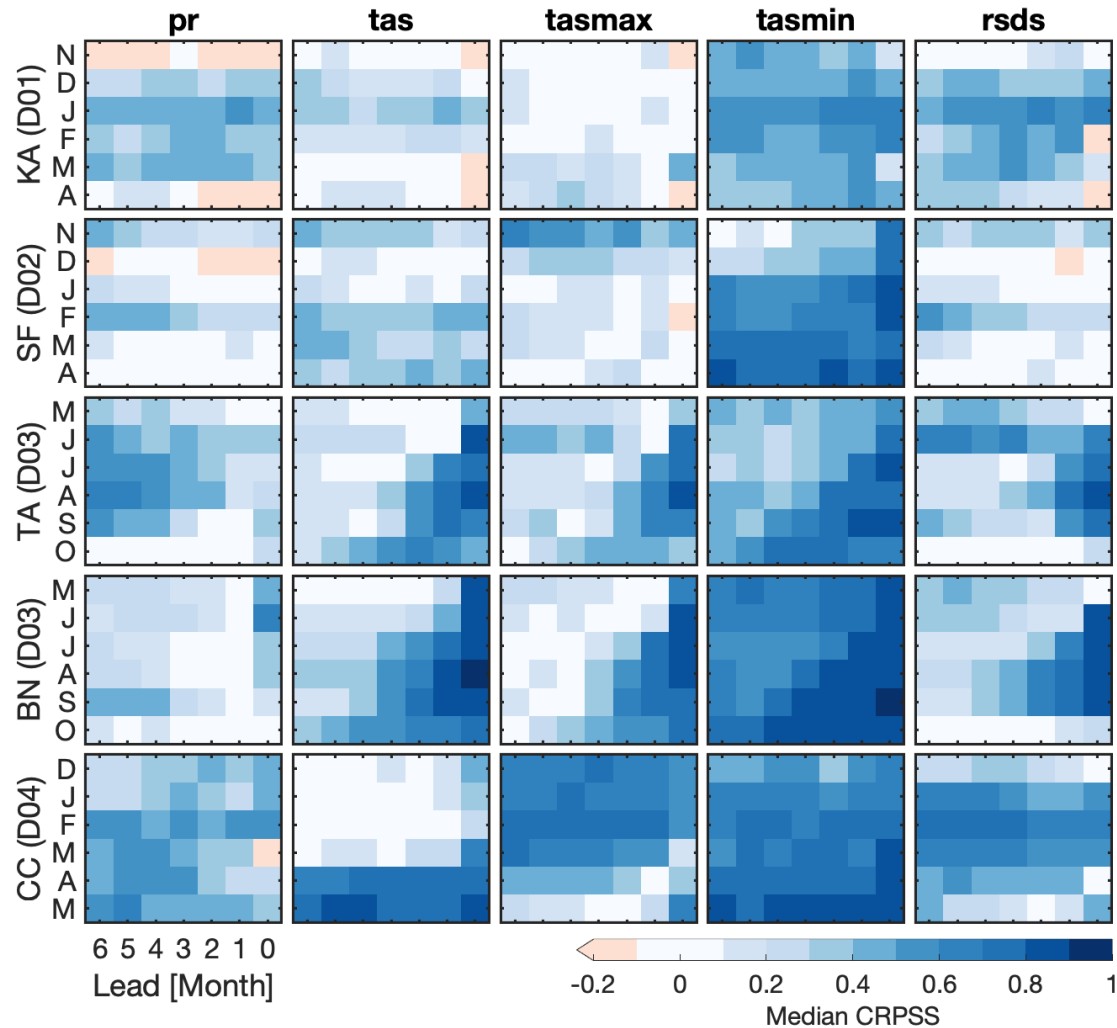

**Figure 8.** Median Continuous Ranked Probability Skill Score (CRPSS) of area averages over the five basins (from top to bottom) of SEAS5-BCSD against raw SEAS5 forecasts with respect to the reference ERA5-Land. The CRPSS values are derived for precipitation (pr), mean (tas), maximum (tasmax), and minimum (tasmin) temperature and shortwave radiation (rsds) as monthly medians for each of the six months during the wet season (x-axis) of the period 1981 to 2016 for each lead-time (y-axis) separately. Blueish (reddish) colors indicate better (worse) correspondence with ERA5-Land after applying the BCSD to the SEAS5 forecasts.

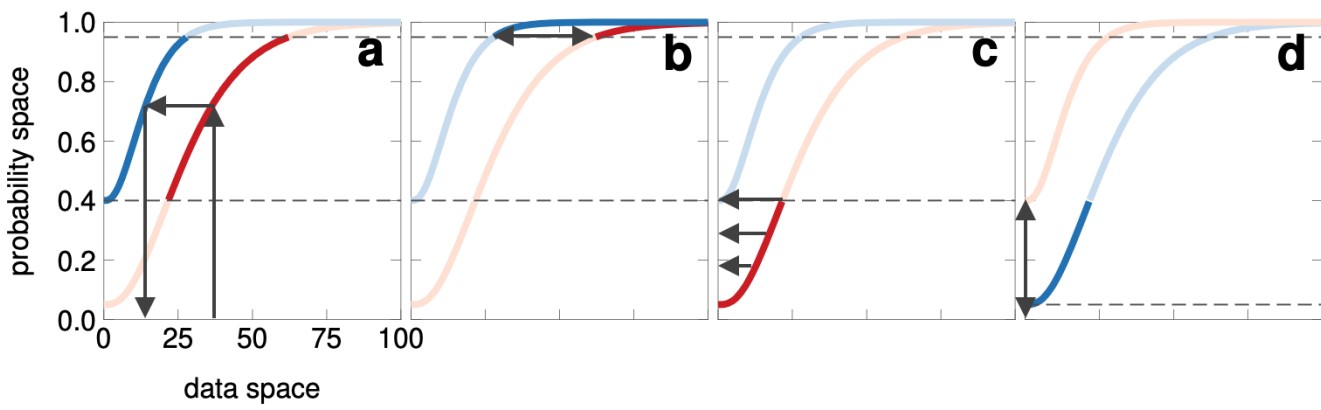

**Figure A1.** a) Empirical quantile mapping between model-based (red) and reference (blue) data; b) delta-approach for correcting extreme values above the maximum Weibull quantiles; c) correction of precipitation intermittency when the dry-day probability of the reference (lower dashed line) is higher; d) correction of precipitation intermittency when the dry-day probability of the reference is lower