# Peer review of "Bias-corrected and spatially disaggregated seasonal forecasts: a long-term reference forecast product for the water sector in semi-arid regions"

_Earth System Science Data, 2020_

## Referee Comment (RC1) · Anonymous Referee #1 · 1 Dec 2020

The authors implemented and evaluated the performance of an bias-correction and spatial-disaggregation (BCSD) approach to seasonal precipitation, temperature and radiation forecasts of the latest long-range seasonal forecasting system SEAS5/ECMWF. The method was applied in four different semi arid basins of the World: the Karun (Iran), the São Francisco (Brazil), the Tekeze-Atbara and Blue Nile (Sudan, Ethiopia and Eritrea), and the Catamayo-Chira (Ecuador and Peru).

The proposed approach was compared to the ERA5-Land/ECMWF and outperformed it in terms of spatial resolution (from 36 km to 0.1°) and spatial patterns agreement.

[Figure]

Also, according to their results, it remarkably reduced lead-dependent drift effects. It would be important to have an idea of the proposed approach relative performance to systems that are available for those regions, but I recognize the amount of work this would demand. Thus, I only suggest the authors to include in their paper a brief comment on the information available to water managers in these four regions. I commend the authors to made freely available the SEAS5 BCSD forecasts (from 1981 to 2019) to the public through the World Data Center for Climate (WDCC), which is hosted by the German Climate Computing Center (DKRZ) in Hamburg, Germany.

My main concern about this paper is not on the method itself, since that was clearly demonstrated its improved performance relative to the competing method, but it is on the raised constraints to the usefulness of seasonal forecasts, particularly in developing countries. The authors point out that there are, based on the literature, different reasons for the effectiveness usefulness, among them: 1. proper communication and application of these forecasts (White et al., 2017); 2. credibility, legitimacy, scale, cognitive capacity, procedural and institutional barriers, and available choices (Patt and Gwata, 2002).

However, the problem goes far beyond these issues: 1. Too much emphasis on the infrastructure solution, which overshadows the importance of preparedness, for example, contingency plans for specific sectors. The focus on developing countries is on the increase of the water supply, but little, or none, effort is undertaken on demand management; 2. There is an institutional challenge in terms of the need for more collaboration among institutions, in particular, when they belong to different levels of administration. Most institutions operate the same way when they were created and they have to face new challenges (environment, society, ...); 3. The water management system does not reach the local level, even this impacting the large management systems. In some regions the density of small (unmonitored) dams is of the order of 0.6 dams/km2. At this scale, farmers use water as long as it is available. When water is no longer available, they look for new sources. There is an urgent need for rethinking

the water governance at this level: more engagement of municipalities and local communities is necessary. In my opinion, the key for disaster preparedness and adaptation is governance at local level, in particular, in dealing with extreme events.

I would add to this list that is key to understand the decision making process for these basins: What is the decision calendar in these basins? What decisions are made and on what basis? What information has the potential to be used for the studied basins (depending on the water system, the interest in the forecast is specific)? How could the information produced be incorporated? Another point, is the forecast issued in a moment compatible with this decision calendar (in some systems this is simply not possible*)? It would be important to include a discussion on these points for these basins. In my view, the promise of the usefulness of seasonal forecasts has been largely due to not trying to answer these questions before designing the information system based on seasonal forecasts.

In my view the topic is of interest of reader of ESSD and the paper does represent a significant contribution for this journal. However, since the authors highlighted the constraints in the effective usefulness of seasonal forecasts, I stress the importance in introducing some discussion on the points raised by this reviewer. —- *Note: It may be necessary the combination of scenario drawing in the moment the decisions are made and revisit such decisions in the moment the climate forecast system can provide useful information to the water sector.

---

## Referee Comment (RC2) · Anonymous Referee #2 · 21 Jan 2021

The authors apply BCSD (actually SD first then BC) to ensemble seasonal forecasts from ECMWF, for five basins in four arid regions. They use ERA5-Land (hourly) as reference data but that in itself represents a down-scaled (time and space replay of ERA5 atmosphere to global land) product. They make numerous references to the importance of topography but then give it almost no attention in results.

I do not think this fits in ESSD. Nothing about ESSD handling or not handling model products. Instead a fundamentally different approach to error terms and uncertainties. A forecast has some skill realized against actual outcomes: forecast 20 mm of rain in a given future period, validated or not against measured rainfall (with spatial and measurement errors!) during that forecast period. Some weather services extract probabilities from their ensemble forecasts, e.g 50% chance of rain or snow, combined with some publicly acknowledged uncertainty of amounts, e.g. up to 3 cm of rain or snow expected, for shorter-term forecasts. I accept that seasonal forecasts present different challenges. Here, however, authors treat the forecasts as perfect (= certain) and likewise the reanalyses as certain and then, despite having introduced substantial but unspecified additional uncertainty by downscaling to 10 km and hourly, spend their efforts trying close gaps between forecasts and higher-resolution reanalyses. Nothing wrong with their approach, but ESSD focuses explicitly and extensively on real-world uncertainties (e.g. read 'uncertainty' paragraphs in ESSD guidelines at https://www.earth-syst-sci-data.net/10/2275/2018/). A typical ESSD paper describes uncertainties of a measurement (e.g. PM2.5 in Christchurch) in terms of instrument errors, measurement errors, operational errors, etc. Then and only then would one attempt to calculate uncertainty of an air quality forecast. 'Uncertainty' is a different problem for ECMWF and for these authors than in most ESSD papers. That difference causes the mismatch? In review that follows I express the view that authors tend to over-sell their product but I do not doubt their motivation or their skill. I doubt that their description belongs in ESSD.

Page 1 line 19 and following: Domain numbers e.g. DO4 come from ECMWF forecasts, from DKRZ labelling, or for author convenience? Used extensively in some sections of results and figures but in other places authors seem to rely more on geographic acronyms e.g. CC-basin. Use / need both?

Page 3 line 29: "huge" another press opinion or outcome of a peer-reviewed study?

Page 3 line 30: "urgent need" expressed by who? The authors?

Page 4 lines 8 to 11: previous limitations mostly applied to 'short-term' not 'seasonal' forecasts. The authors make very high claims for this product without any evidence.

Page 4 line 14: what does 'reference' mean in this sentence?

Page 4 line 20: 5 days before the present?

Page 4 line 25 to 29: this text comes almost verbatim from the landing page for ERA5-Land. Authors should cite that?

Page 5 Table 1: Nothing about elevation or topographic complexity of basins. Earlier, authors listed elevation corrections as a necessary or desirable feature?

Page 5 line 11: bias correcting to what?

Page 5 line 15: readers will likely know forecast skill score but the term "highest" conveys nothing about skill level?

Page 6 line 6: "crucial' to understand orography but authors give only generalities ("up to 4000 m" Fig 1 not much help, only color-coded 2-D. Give us an elevation profile for stream level 1?

Page 6 line 9: no doubt, but by who's definition? Or what reference?

Page 6 line 10: "dangerous"?

Page 6 line 12: "assumed to experience an increase in the frequency and severity" assume by who, what references. Likely true but on what basis? References that follow in this paragraph document past extreme events but largely avoid prediction?

Page 7 line 8: "these anomalies" - the remaining differences between forecast and reference data once the climatological mean reference has been subtracted?

Page 8 line 9: "fairly large number of samples for both the reference" but these represent data sparse regions?

Page 9 Model Biases: extensive discussion of how the uncorrected forecasts fail but why do we care? Useful discussion starts at line 24?

Page 9 line 29: "RMSE of SEAS5 BCSD is much lower compared to the raw forecasts." Strong statement not supported by Figure 3. This statement from line 33 "other cases where the bias-correction shows almost no improvement" seems more accurate. For this reader, Fig 3 shows that when RMSE differences occur, they generally favor the BCSD product, while in other cases one can not distinguish RMSE terms between raw and corrected. We also need, as the authors hint but do not show, some uncertainty limits here? All precip RMSE, except for one station, lie below 2 mm/day, often below 1 mm/day. Do the authors claim such accuracy in their base numbers? One doubts. For tas, again except for 1 station, essentially all RMSE lie below 1k. The authors expect us to believe with their tools they can distinguish products at 2 mm/day and 1k? Remarkable if true but they give us no evidence. A low correlation error (RMSE) between two products of assumed 'perfection' but almost certainly with high inherent fundamental uncertainties seems of little relevance?

Page 10: Reader needs to jump from Fig 3 in 4.1 to Fig 6 in 4.2 then back to Fig 4 in 4.3. Reason for this hopping around? Hopping will disappear once Figures take their appropriate place in final document but then sequence will look wrong?

Page 10 Section 4.2 resolution: no uncertainties here? These are average sums of 4-month periods from 25 to 51 ensemble runs over 35 years. They must have SD, 95CI, etc? Almost every number and result across the manuscript has substantial uncertainty ranges but authors treat everything as exact?

Page 10 section 4.3 lead-time: without ranges or uncertainties, reader has no basis to accept any of these supposed differences or patterns.

Page 10 line 20: weather patterns may shift but locations do not shift, southward or any other direction

Page 11 line 3: reader needs to go from Fig 4 in previous paragraph now to Fig 7. Consider a more helpful and logical sequence??

Page 11 line 5: reader now moves from geographic codes KA or CC back to domain codes D03. Why? Confusing!

Page 11 section 4.5 overall skill: many readers will know these skill scores but will usually have seen them expressed as a range. This reader has no confidence in an absolute CRPSS of 0.4 but might accept a range from 0.3 to 0.5? Again, authors treat their results as absolute when in fact they contain substantial uncertainty!

Page 11 section 5 Discussion: helpful discussion of regional factors follows, intended apparently as justification for why corrected products seem occasionally but not consistently to outperform original forecasts. Very real regional challenges, no doubt. But if the original forecast products lacked sufficient skill when confronted by meteorological and topographic details of each basin, bias correction to higher resolution will not remove that fundamental detail-driven uncertainty? It may raise skill scores but still miss key local details. E.g. it will continue to show high fundamental uncertainty! Vis "spatial and temporal inconsistencies in the forecasted spatial extent and intensity" (Page 12 line 7) of precip, of temperature, of clouds, etc. represent the real-world uncertainty not included and certainly not overcome! The authors themselves make this point (Page 12 line 13) that for basins with skill score improvements of 0 and no differences in RMSE, fundamental uncertainty has defeated their good efforts!

Page 27 Figure 2: These are composite biases (areal sum of daily data) for source forecast vs ERA5-Land reanalysis? The colours - almost impossible to distinguish even in the label) represent different lead times from 0 to 11 months? Or are these monthly averages? Not clear. After working extensively similar Fig 3, I still find these graphics difficult to read and interpret.

At this point this reader largely 'gave up'. The data description for DKRZ seems easy to use and very helpful. Authors have provided useful guidance to static products and how to find updates. Generally ESSD does not allow: 'contact the author' (Page 15 line 23). Appendices provide useful documentation on BC, on error calculations, and on skill scores. Overall the authors have provided useful information. Their approach however still seems orthogonal to the intent of ESSD.

---

## Author Response (AR1)

**Author Response for ESSD-2020-177**

First of all, we would like to thank both reviewers and the editor for the fruitful and helpful comments and discussions. In our revised manuscript, we tried to address the raised concerns, points of criticism and corrections. Please find below the comments from both reviewers as well as our reply and the corresponding changes in the manuscript.

Some of the major changes in the revised version are as follows. We have
- included an overview of current seasonal forecasting initiatives, as suggested by reviewer 1
- included some aspects about further technical and societal requirements for ensuring a transfer of current seasonal forecasting products into practice, as suggested by reviewer 1
- enhanced the description of „negative" results (I.e. where the bias-correction did not improve or even worsen the raw seasonal forecasts), as suggested by reviewer 2
- underlined some of the statements e.g., about the Grand Ethiopian Renaissance Dam with references, as suggested by reviewer 2
- included some measures for describing the variations in the reference data and our forecasts, as suggested by reviewer 2
- and added more information about the river basins in the four study domains.

Besides that, we revised the order of figures in the manuscript, changed the colours in Figure 2 and 3 for improving the readability and changed the wording in several parts.

If there were changes in the manuscript based on the comments of a reviewer, we have marked them here in this reply with their page- and line-numbers in the manuscript with tracked changes.

**Response to Reviewer #1**

**General comments**

*The authors implemented and evaluated the performance of a bias-correction and spatial-disaggregation (BCSD) approach to seasonal precipitation, temperature and radiation forecasts of the latest long-range seasonal forecasting system SEAS5/ECMWF. The method was applied in four different semi-arid basins of the World: the Karun (Iran), the São Francisco (Brazil), the Tekeze-Atbara and Blue Nile (Sudan, Ethiopia and Eritrea), and the Catamayo-Chira (Ecuador and Peru).*

*The proposed approach was compared to the ERA5-Land/ECMWF and outperformed it in terms of spatial resolution (from 36 km to 0.1◦) and spatial patterns agreement. Also, according to their results, it remarkably reduced lead-dependent drift effects. It would be important to have an idea of the proposed approach relative performance to systems that are available for those regions, but I recognize the amount of work this would demand. Thus, I only suggest the authors to include in their paper a brief comment on the information available to water managers in these four regions. I commend the authors to made freely available the SEAS5 BCSD forecasts (from 1981 to 2019) to the public through the World Data Center for Climate (WDCC), which is hosted by the German Climate Computing Center (DKRZ) in Hamburg, Germany.*

*My main concern about this paper is not on the method itself, since that was clearly demonstrated its improved performance relative to the competing method, but it is on the raised constraints to the usefulness of seasonal forecasts, particularly in developing countries. The authors point out*

*that there are, based on the literature, different reasons for the effectiveness usefulness, among them: 1. proper communication and application of these forecasts (White et al., 2017); 2. credibility, legitimacy, scale, cognitive capacity, procedural and institutional barriers, and available choices (Patt and Gwata, 2002).*

*However, the problem goes far beyond these issues:*

1. *Too much emphasis on the infrastructure solution, which overshadows the importance of preparedness, for example, contingency plans for specific sectors. The focus on developing countries is on the increase of the water supply, but little, or none, effort is undertaken on demand management;*
2. *There is an institutional challenge in terms of the need for more collaboration among institutions, in particular, when they belong to different levels of administration. Most institutions operate the same way when they were created and they have to face new challenges (environment, society, …);*
3. *The water management system does not reach the local level, even this impacting the large management systems. In some regions the density of small (unmonitored) dams is of the order of 0.6 dams/km2. At this scale, farmers use water as long as it is available. When water is no longer available, they look for new sources. There is an urgent need for rethinking the water governance at this level: more engagement of municipalities and local communities is necessary. In my opinion, the key for disaster preparedness and adaptation is governance at local level, in particular, in dealing with extreme events.*

*I would add to this list that is key to understand the decision-making process for these basins: What is the decision calendar in these basins? What decisions are made and on what basis? What information has the potential to be used for the studied basins (depending on the water system, the interest in the forecast is specific)? How could the information produced be incorporated? Another point, is the forecast issued in a moment compatible with this decision calendar (in some systems this is simply not possible\*)? It would be important to include a discussion on these points for these basins. In my view, the promise of the usefulness of seasonal forecasts has been largely due to not trying to answer these questions before designing the information system based on seasonal forecasts.*

*In my view the topic is of interest of reader of ESSD and the paper does represent a significant contribution for this journal. However, since the authors highlighted the constraints in the effective usefulness of seasonal forecasts, I stress the importance in introducing some discussion on the points raised by this reviewer.*

*\*Note: It may be necessary the combination of scenario drawing in the moment the decisions are made and revisit such decisions in the moment the climate forecast system can provide useful information to the water sector.*

**Reply:** We would like to thank the reviewer for the generally positive feedback for our study. Furthermore, we highly appreciate the constructive and thoughtful comments about the usage and transfer of seasonal forecasts into practice.

First of all, we would like to acknowledge the reviewer's comment that we should at least mention similar products and initiatives in our manuscript. We fully agree and will add such a list including global initiatives like the WMO Long-Range Forecast Multi-Model Ensemble (https://www.wmol-

[c.org](c.org)), the North American Multi-Model-Ensemble (NMME, [https://www.cpc.ncep.noaa.gov/products/NMME/](https://www.cpc.ncep.noaa.gov/products/NMME/)), the C3S Seasonal Forecasts ([https://climate.copernicus.eu/seasonal-forecasts](https://climate.copernicus.eu/seasonal-forecasts)), and the International Research Institute for Climate and Society (IRI, [https://iri.columbia.edu/our-expertise/climate/forecasts/seasonal-climate-forecasts/](https://iri.columbia.edu/our-expertise/climate/forecasts/seasonal-climate-forecasts/)) as well as regional initiatives like the forecasts from the IGAD Climate Prediction and Application Centre (ICPAC, [https://www.icpac.net/seasonal-forecast/](https://www.icpac.net/seasonal-forecast/)) or the EURO-Brazilian Initiative for improving South American seasonal forecasts (EUROBRISA, [http://eurobrisa.cptec.inpe.br](http://eurobrisa.cptec.inpe.br)) and a short discussion to our paper. With respect to a quantitative comparison of our forecasts with such products, we have to state that this is extremely difficult as particularly ensemble-based categorical forecast highly depend on several fundamental aspects (e.g., which "baseline-period" and reference products were used for defining the climatology? which thresholds were used for defining categories? how were the forecasts from different issue dates combined?). Thus, we should rather aim at a qualitative comparison (e.g., did both systems predict a dry or wet month? what was the probability of > 300mm of rainfall?). This, however, would be a comprehensive study on its own and is something that we are already looking into.

Furthermore, we also agree that there are many other issues with respect to the usefulness of seasonal and longer-term forecasts particularly in developing countries. But, at the same time, we must state that finding solutions for these issues are far beyond the scope of this study as this is first and foremost a scientific publication about a dataset and, hence, would not be the right place to discuss fundamental challenges in the practice transfer of seasonal forecasts.

Especially the three additional issues that the reviewer defines require substantial societal and administrative reorganization of the water sector. We have also experienced conflicts between authorities and institutions in our target regions by ourselves, which often make a direct and efficient collaboration difficult. Furthermore, with respect to a sustainable transfer of such forecasts into practice, we would have to put a lot of effort in the education and coordination of potential end-users of such information as well as in the definition of well-coordinated action plans, that are approved by various local stakeholders.

All these challenges cannot be addressed in such a technical manuscript. However, one aspect, that was communicated during the various meetings we had in the target regions, is the lack of tailored regional and freely available seasonal forecasts as well as an introduction in the handling with such ensemble-based information. While there are several global products available, most of these products are "raw" forecasts and still require a lot of post-processing in order to fulfill the demands allowing to serve as a decision-support for local water management. Due to the lack of computational resources, an insufficient experience with the treatment of large ensemble forecasts, a limited bandwidth for the download, and other reasons, this post-processing is often a major obstacle for many institutions in developing countries.

Hence, this particular step was done in this study by obtaining a long period of global re-forecasts from ECMWF and applying a bias-correction and spatial disaggregation for improving the spatial resolution and making the forecasts consistent with a state-of-the-art reference product. Moreover, the SEAS5-BCSD-forecasts, can be (and already are) freely accessed and used directly for deriving probabilistic forecasts for e.g., extreme warm or wet conditions and other forecast quantities, which are required for the day-to-day water management. In that sense, we think that our and similar products are an important contribution towards an improved governance of the water sector in developing countries.

The reviewer also mentions that any newly developed decision-making system has to take the decision-making process in the basins into account. Again, we completely agree with these points and can confirm that regionalized forecast quantities (e.g., drought indicators, categorical forecasts, etc.) have to be consistent with local conditions and needs. And these requirements can only be identified in consultation and close iteration with local water experts.

We also acknowledge that there is a gap between the scientific developments in seasonal and longer-term forecasting during the recent years and the efforts to bring this information to authorities and institutions particularly in developing countries, where such forecasts could be crucial for an improved and more sustainable water management. We therefore hope that our dataset and publication are a small step for overcoming this gap.

To conclude, we fully agree with the concerns raised by the reviewer. As these are important challenges that have to be addressed for ensuring a successful transfer of such newly developed products into practice, we will include a dedicated part in the discussion.

**Changes:** We have included a list of current seasonal forecasting initiatives and projects in the introduction (page 2, lines 14-22). Furthermore, we have added some of the mandatory requirements for a successful transfer of seasonal forecasts into practice to the conclusion (page 16, lines 11-25).

**Response to Reviewer #2**

**General comments**

*The authors apply BCSD (actually SD first then BC) to ensemble seasonal forecasts from ECMWF, for five basins in four arid regions. They use ERA5-Land (hourly) as reference data but that in itself represents a down-scaled (time and space replay of ERA5 atmosphere to global land) product. They make numerous references to the importance of topography but then give it almost no attention in results.*
*I do not think this fits in ESSD. Nothing about ESSD handling or not handling model products. Instead a fundamentally different approach to error terms and uncertainties. A forecast has some skill realized against actual outcomes: forecast 20 mm of rain in a given future period, validated or not against measured rainfall (with spatial and measurement errors!) during that forecast period. Some weather services extract probabilities from their ensemble forecasts, e.g 50% chance of rain or snow, combined with some publicly acknowledged uncertainty of amounts, e.g. up to 3 cm of rain or snow expected, for shorter-term forecasts. I accept that seasonal forecasts present different challenges. Here, however, authors treat the forecasts as perfect (= certain) and likewise the reanalyses as certain and then, despite having introduced substantial but unspecified additional uncertainty by downscaling to 10 km and hourly, spend their efforts trying close gaps between forecasts and higher-resolution reanalyses. Nothing wrong with their approach, but ESSD focuses explicitly and extensively on real-world uncertainties (e.g. read 'uncertainty' paragraphs in ESSD guidelines at https://www.earth-syst-sci-data.net/10/2275/2018/). A typical ESSD paper describes uncertainties of a measurement (e.g. PM2.5 in Christchurch) in terms of instrument errors, measurement errors, operational errors, etc. Then and only then would one attempt to calculate uncertainty of an air quality forecast. 'Uncertainty' is a different problem for ECMWF and for these authors than in most ESSD papers. That difference causes the mismatch. In review that follows I express the view*

*that authors tend to over-sell their product but I do not doubt their motivation or their skill. I doubt that their description belongs in ESSD.*

**Reply:** We would like to thank the reviewer for the feedback. Before we start with a detailed reply to each of the comments, we would like to give a general statement about this particular review. While we highly appreciate several constructive comments that actually led to an improvement of the manuscript, we feel that two of the main points of criticism cannot be addressed in a way that the reviewer will be fully satisfied:

1. According to the reviewer, our manuscript does not fit into the scope of ESSD. This is mentioned several times: "*I do not think this fit in ESSD*", "*I doubt that their description belongs in ESSD*", "*Their approach however still seems orthogonal to the intent of ESSD*"

2. The reason for this is that, according to the reviewer, we did not provide a full-fledged uncertainty analysis: "*but ESSD focuses explicitly and extensively on real-world uncertainties*", "*`Uncertainty' is a different problem for ECMWF and for these authors than in most ESSD papers*".

Prior to submitting our manuscript to ESSD, we have approached the editor and discussed, if our paper fits into the scope of the journal. For this purpose, we have also submitted an extended abstract which included the key aspects of our study and dataset. It was discussed and, finally, concluded and agreed that a publication in ESSD is justified mainly due to two reasons:

- Obtaining reliable and consistent observation-based long-term and high-resolution information in our study regions is almost impossible (as also acknowledged by the reviewer): A decreasing number of stations used in global station-based products (Lorenz and Kunstmann, 2012, Lorenz et al., 2014) and a lack of continuous local station data in these regions limit the options for reliable reference data. However, information about incoming water resources as well as their long-term trends and dynamics are crucial for the sustainable water management in such climatically vulnerable dry regions. Due to this dilemma, we cannot rely solely on incomplete observations, but need to expand the data sources, e.g., to model-based information.

- Our used reference dataset ERA5-Land is a model-based reanalysis product. Despite no direct usage of observations in the production of ERA5-Land as an offline re-run of ECMWFs latest reanalysis ERA5, it benefits from the millions of observations that have been assimilated in the ERA5 atmospheric forcing as well as from the lapse rate correction of input air temperature, air humidity and pressure in the interpolation step to consider the importance of topography at the higher resolution. ERA5-Land should therewith ensure a high quality and high resolution information of surface variables.

Thus, the need for reference alternatives for our study regions required the use of state-of-the-art model-based high-resolution reanalyses. Single hydrometeorological variables such as precipitation could also have been provided by high-resolution remote sensing-based information, but the required consistency and intrinsic dependence structure of all considered variables for subsequent impact modeling could not be provided by using several different datasets, most likely also at different spatial resolutions. Moreover, the design of our framework allows us to easily extend the set of forecasted variables and domains as both ERA5-Land and SEAS5 provide a wide range of consistently defined global hydrometeorological variables. We therefore demonstrated a sound solution, imposed by the constraints, for securing a reference dataset in data-sparse regions to be able to finally provide improved bias-corrected regionalized seasonal forecasts for decision-support and impact modeling. That being said, we have put tremendous efforts in the evaluation and treatment of uncertainties in previous studies (see e.g., Lorenz and Kunstmann, 2012; Lorenz et al., 2014; Sneeuw et al., 2014; Lorenz et al. 2015). So, while we fully understand the criticism of the reviewer, we hope that this discussion helps to comprehend the design of our approach.

As the reviewer raises several points of criticism in the general comments, we would like to answer to each of the raised issues point by point.

*They make numerous references to the importance of topography but then give it almost no attention in results.*
**Reply:** ERA5-Land is based on the spatial downscaling of ERA5. This downscaling also includes a thermodynamic orographic adjustment (see e.g. the presentations from Muñoz Sabaters et al. 2017, 2018 or the landing page for ERA5-Land, ECMWF, 2019). So, while we do not apply an "explicit" orographic adjustment, we use a reference dataset which was corrected for orography. This also means that by applying a bias-correction towards ERA5-Land, we automatically include an implicit orographic adjustment.

*I do not think this fits in ESSD. Nothing about ESSD handling or not handling model products. Instead, a fundamentally different approach to error terms and uncertainties.*
**Reply:** See our general comments and discussion above. Suitability has been confirmed prior to the submission by the editorial board.

*A forecast has some skill realized against actual outcomes: forecast 20 mm of rain in a given future period, validated or not against measured rainfall (with spatial and measurement errors!) during that forecast period.*
**Reply:** As the reviewer also mentions in a later comment, we are looking at data-sparse and orographically complex regions. While we also made comparisons against the (very few) station-based observations in preparatory studies, we think that comparing a model-based product with a spatial resolution of 10 km against point-based measurements in such complex domains only allows for limited insights. Moreover, in such regions, evaluating global data is always a compromise as you can either use few station-based observations (which come with their own shortcomings and issues) or rely on gridded reference products like, e.g., ERA5, ERA5-Land or (for rainfall) more specific datasets like MSWEP or CHIRPS. That being said, in the submitted article we have already included a comparison of the forecasts against actual outcomes in Figures 2 and 3: We show the Bias and Root Mean Squared Error of SEAS5-BCSD and SEAS5 against our reference ERA5-Land. It should be further noted that the aim of any bias-correction is to make a forecast more consistent with a reference product, and not necessarily the improvement of the prediction skill, which is something totally different (see also our reply to the reviewer's comment to *Page 11 section 5).*

*Some weather services extract probabilities from their ensemble forecasts, e.g., 50% chance of rain or snow, combined with some publicly acknowledged uncertainty of amounts, e.g., up to 3 cm of rain or snow expected, for shorter-term forecasts.*
**Reply:** This is totally true but such *public* information requires a lot of preliminary groundwork and this is exactly the main purpose of our dataset. For deriving, e.g., probabilities for rain and snow, you need to introduce deterministic thresholds. This, however, is a problem particularly for longerterm forecasts due to the model drift. As an example, 3mm/day of rainfall can correspond to the 10%-quantile during lead 0, while it corresponds to the 30%-quantile during higher leads. If such probabilistic information (50% chance of rain and snow) should be derived from the forecasts, one needs to correct for these drifts and this is one of the outcomes from our study. Furthermore, information about the uncertainty (or spread) of e.g., up to 3mm is useless if the climatology and natural variability of rainfall is not taken into account. An ensemble spread of 3mm in a dry region indicates a highly unsharp forecast while we would not care if such values are obtained over e.g., high-precipitation monsoon regions. This shows that the forecast information, that we're used to obtain from weather services, requires a) a reliable and consistent (w.r.t. some kind of reference data) re-forecast product over a quite long period to be able to correct for biases and b) some understanding about the local climate conditions.

During our joint workshops and meetings in the target regions, it was clearly stated by local authorities, researchers, and stakeholders that there is currently a lack of tailored regional seasonal forecast systems in almost all our study regions and we are convinced that our dataset is a promising contribution for developing such systems in the future. To conclude, our dataset serves exactly this purpose: that weather services, stakeholders and other water experts in the study regions are enabled to apply regionalized seasonal forecasts.

*Here, however, authors treat the forecasts as perfect (= certain) and likewise the reanalyses as certain…*

**Reply:** While we acknowledge that reanalysis products are far from perfect (Lorenz et al., 2012, Lorenz et al. 2014, Gleixner et al. 2020), they already include millions of observations and, moreover, are often the only source of consistent hydrometeorological information in data-sparse regions (Gleixner et al., 2020). This was already stated in our introduction (page 3, line 25). But besides these concerns, recent studies already certify a performance of state-of-the-art reanalyses that is similar to those from observation-based datasets (see e.g. Tarek et al., 2020). With respect to the forecasts, we included the comparison of the wet-day-probability (Figure 7) and the CRPSS (Figure 8), which both take the whole ensemble and it's spread into account, still demonstrating the "uncertainty" of the improved forecasts.

*…and then, despite having introduced substantial but unspecified additional uncertainty by downscaling to 10 km and hourly, spend their efforts trying close gaps between forecasts and higher-resolution reanalyses.*

**Reply:** We agree that any downscaling approach can introduce additional uncertainty. But it is not the scope of this publication to perform an error propagation for a classical bilinear interpolation. Furthermore, we perform no temporal downscaling as both the reference and forecast data are available at daily resolution (i.e., there is no hourly data used in our study). We also do not understand why the reviewer is complaining that we are trying to close gaps between forecasts and higher resolution reanalyses as this is exactly the aim of any downscaling approach.

*Nothing wrong with their approach, but ESSD focuses explicitly and extensively on real-world uncertainties (e.g. read 'uncertainty' paragraphs in ESSD guidelines at https://www.earth-syst-sci-data.net/10/2275/2018/). A typical ESSD paper describes uncertainties of a measurement (e.g. PM2.5 in Christchurch) in terms of instrument errors, measurement errors, operational errors, etc. Then and only then would one attempt to calculate uncertainty of an air quality forecast.*

**Reply:** We have difficulties understanding the "real-world uncertainties" mentioned by the reviewer. How can we obtain such "real-world uncertainties" if the "true" state in such regions is unknown or only accessible at some very few locations? Particularly in data-sparse regions, we have

limited knowledge and data which makes it almost impossible to quantitatively validate a distributed model at every single location. So, every evaluation is relative as we always have to refer to some reference (reanalysis, remote sensing products, etc.), which is often far from perfect. Regarding the uncertainty of the improvement of the forecasts to our chosen reference product ERA5-Land, we provide the CRPSS (Figure 8).

The reviewer is further referring to a full-fledged error propagation from the measurement through the whole assimilation in a reanalysis product (which is used for initializing a forecast) down to the final forecasted variable. While we fully acknowledge that this propagation is crucial for purely observation-based datasets, it is impossible to realize in such a complex model-cascade.

*In review that follows I express the view that authors tend to over-sell their product but I do not doubt their motivation or their skill.*

**Reply:** We do not want to raise the impression that we're over-selling our product. Despite the fact that this is one of the first publicly available regional seasonal forecast products that also provides operational forecasts, we show in several figures and analyses how our framework improves the raw forecasts. Besides this, we also mention shortcomings of the approach already in the abstract (page 1, lines 11 – 13) and extensively discuss these limitations on page 13, lines 4 – 8 or page 14, lines 9 – 14.

**Minor comments**

*Page 1 line 19 and following: Domain numbers e.g. DO4 come from ECMWF forecasts, from DKRZ labelling, or for author convenience? Used extensively in some sections of results and figures but in other places authors seem to rely more on geographic acronyms e.g. CC-basin. Use / need both?*

**Reply:** We have decided to use domain numbers that can be easily expanded. This is why we have enumerated the study areas in our manuscript from D01 (Karun basin, Iran) to D04 (Catamayo-Chira basin, Ecuador / Peru). These numbers have been defined in the SaWaM-project (https://grow-sawam.org) in which this study has been conducted. Please note that here, we refer to *domains* and not *basins*. As we've also included an evaluation of basin-averaged forecasts, we also needed some abbreviations for these regions (like, e.g., the CC-basin). Moreover, the third domain D03 actually contains two basins, namely the Blue-Nile-basin (BN) and the Tekeze-Atbara-basin (TA). We therefore need both the domain numbers and the basin acronyms. This distinction will be clarified in the revised manuscript.

**Changes:** We have clarified the distinction between domains and basins (page 6, lines 1-2). Furthermore, we have added some more details about the different study regions (including some topographic aspects, page 6, line 6 to page 7, line 14).

*Page 3 line 29: "huge" another press opinion or outcome of a peer-reviewed study?*

**Reply:** It was stated in many scientific publications that the GERD will have significant implications for the whole Nile Basin (e.g., Wheeler et al. 2020, Basheer et al. 2020). But in order to sound a bit less sensational, we will re-phrase the respective sentence and add references to Kidus et al. (2019), Wheeler et al. (2020) and Basheer et al. (2020).

**Changes:** We have re-phrased the respective part and added some more references (page 4, lines 3-6).

*Page 3 line 30: "urgent need" expressed by who? The authors?*

**Reply:** It was stated in many scientific publications that longer-term forecasts have the potential to significantly improve the regional water management, particularly in water-scarce regions which highly depend on the incoming freshwater resources from the rainy seasons. While multiple examples were already provided in the first part of the introduction, we will re-phrase the respective sentence and add references to Tall et al. (2012) and Gerlitz et al. (2020).

**Changes:** We have re-phrased the respective part and added some more references (page 4, lines 7-10).

*Page 4 lines 8 to 11: previous limitations mostly applied to 'short-term' not 'seasonal' forecasts. The authors make very high claims for this product without any evidence.*

**Reply:** We did not fully grasp the direction in which the reviewer was aiming with the mentioned *limitations*. The limitations of forecasts with different forecast horizons, that can be corrected with post-processing methods, are similar because the underlying model systems are similar. As an example, at ECMWF, most forecasts products and reanalyses are based on a single model system called the Integrated Forecasting System (IFS). Similarly, other atmospheric model systems like the Weather Research and Forecasting Model (WRF) are used for developing short-term forecasts (e.g. Vladimirov et al. 2020) as well seasonal predictions (e.g. Siegmund et al. 2017) and climate simulations (e.g. Heinzeller et al. 2018). Thus, issues like a low spatial resolution, model biases or model drifts are not due to a specific forecast horizon, but rather due to the general usage of outputs from global hydrometeorological models.

If the reviewer is referring to the six limitations that were defined by Patt and Gwata (2002), it should be noted that this reference was explicitly about the usage of seasonal forecasts, as already mentioned in the title (*Effective seasonal climate forecast applications: examining constraints for subsistence farmers in Zimbabwe*).

Furthermore, we do not think that we make "high claims" without any evidence. We show that, compared to the raw forecasts, our SEAS5-BCSD has an improved resolution, reduced biases and, hence, better consistency with ERA5-Land as well as substantially reduced model drifts. Furthermore, we have published and thereby made transparent the whole repository via the DKRZ, so it can be used freely for evaluating the potential of seasonal forecasts in the study regions and for educating local experts.

*Page 4 line 14: what does 'reference' mean in this sentence?*

**Reply:** By the very nature of any bias-correction, we need some reference information towards which we correct the forecasts. This holds true for forecasts on all temporal scales. In our study, we're using data from ERA5-Land as reference information, towards we correct the seasonal forecasts. As we've already mentioned in the manuscript, we are well aware that such products have their limitation but they are often the only source of consistent hydrometeorological information in such data-scarce regions.

*Page 4 line 20: 5 days before the present?*

**Reply:** We of course meant "before" instead of "after". Thank you for this note.

**Changes:** We have corrected the wrong wording (page 4, line 32).

*Page 4 line 25 to 29: this text comes almost verbatim from the landing page for ERA5-Land. Authors should cite that?*

**Reply:** This is true. Thank you for this comment. We will rephrase and add a reference to the respective pages.

**Changes:** We have added a reference to ERA5-Land (page 5, line 9).

*Page 5 Table 1: Nothing about elevation or topographic complexity of basins. Earlier, authors listed elevation corrections as a necessary or desirable feature?*

**Reply:** We agree that we have over-emphasized the topography-aspect in our manuscript. As we only apply an "indirect" topographic correction through the bias-correction towards ERA5-Land, we will re-phrase the respective parts and clarify that we do not apply any further adjustment or dedicated evaluation. Nevertheless, we will include more details about the topography in the revised manuscript.

**Changes:** We have added some discussion about the „indirect" altitude correction (page 14, lines 25-29) and also extended the description of the study domains (page 6, line 6 to page 7, line 11).

*Page 5 line 11: bias correcting to what?*

**Reply:** They have used the Southeast Asia OBServations (SA-OBS) gridded rainfall product as reference. We'll clarify this in the revised manuscript.

**Changes:** We have added the reference in the manuscript (page 5, lines 20-21).

*Page 5 line 15: readers will likely know forecast skill score but the term "highest' conveys nothing about skill level*

**Reply:** We agree that the term "highest" was misleading in this context. We now use, in accordance with the abstract from Gubler et al. 2019, the term "highest prediction performance" (page 5, line 25).

*Page 6 line 6: "crucial' to understand orography but authors give only generalities ("up to 4000 m" Fig 1 not much help, only color-coded 2-D. Give us an elevation profile for stream level 1?*

**Reply:** See our comment to *Page 5 Table 1.* In addition, what exactly is the reviewer referring to with "elevation profile for stream level 1"? If a cross section of the river streams is meant that would not give additional insight in the context of the study.

*Page 6 line 9: no doubt, but by who's definition? Or what reference?*

**Reply:** In the past, we have made extensive analyses with freely available hydrometeorological datasets. As an example, in Lorenz and Kunstmann (2012) or Lorenz et al. (2014), we have evaluated the number of gauges that usually go into global precipitation datasets or which are available via online data portals like GRDC. Prior to this study, we have also looked at the number of stations in each of the basins, which was constantly decreasing during the last decades. While there are certainly more stations available (e.g., operated by local meteorological organizations), it is often difficult to get access to reliable long-term observational data. As the general reference, we will add Lorenz and Kunstmann (2012) and Lorenz et al. (2014).

**Changes:** During the revision, we have decided that we do not want to refer to Lorenz and Kunstmann (2012) or Lorenz et al. (2014) as both publications focus on global-scale analyses. Unfortunately, there are no dedicated publications about the decrease of in situ stations in our study domains and we think that adding such an analysis to this manuscript would go beyond the scope of this study. But if the editor and/or reviewer suggests to add such an analysis, we can do that in the next round.

*Page 6 line 10: "dangerous"?*

**Reply:** We want to emphasize that there is a lack of *in situ* data in particular those climatically sensitive regions, where a continuous, quality-controlled, and reliable observation of major climatic variables is crucial. To sound a bit less sensational, we will use terms such as "worrying" instead

**Changes:** In the revised manuscript, we're using the term „of particular concern" (page 7, lines 14-15).

*Page 6 line 12: "assumed to experience an increase in the frequency and severity" assume by who, what references. Likely true but on what basis? References that follow in this paragraph document past extreme events but largely avoid prediction?*

**Reply:** This increase in the frequency and severity of extreme events was reported in multiple studies. See Marengo et al. (2012), Torres et al. (2017) or Andrade et al. (2020) for regional assessments and Fischer and Knutti (2014) or Touma et al. (2015) for studies on global trends. There is even a dedicated IPCC special report (Shukla et al. 2019), which focuses (amongst others) on climate change, desertification and land degradation. We agree that we should have added some references which support our statement. This will be done in the revised version.

**Changes:** We've added some more references (page 7, line 17).

*Page 7 line 8: "these anomalies" - the remaining differences between forecast and reference data once the climatological mean reference has been subtracted?*

**Reply:** Exactly. We will clarify this in the revised manuscript.

**Changes:** We've rephrased the respective part (page 8, line 12).

*Page 8 line 9: "fairly large number of samples for both the reference" but these represent data sparse regions?*

**Reply:** The number of samples usually refers to the numbers of values that are used for calculating a statistical distribution. Here, we are using data from ERA5-Land and SEAS5 so we actually have values in each single pixel. Data sparsity refers to the lack of *in situ* stations. So, while our regions can be assumed to be data sparse in terms of station data, we have a large number of daily model-based sample data.

*Page 9 Model Biases: extensive discussion of how the uncorrected forecasts fail but why do we care? Useful discussion starts at line 24?*

**Reply:** Before we start to discuss the impact and performance of the bias-correction, we (and the readers) have to understand the overall characteristics and magnitudes of the model biases and how they vary between the study domains. Only then can we put the quantitative results in a meaningful context. We would hence not assume the discussion to be useless.

*Page 9 line 29: "RMSE of SEAS5 BCSD is much lower compared to the raw forecasts." Strong statement not supported by Figure 3. This statement from line 33 "other cases where the bias- correction shows almost no improvement" seems more accurate. For this reader, Fig 3 shows that when RMSE differences occur, they generally favor the BCSD product, while in other cases one can not distinguish RMSE terms between raw and corrected. We also need, as the authors hint but do not show, some uncertainty limits here? All precip RMSE, except for one station, lie below 2 mm/day, often below 1 mm/day. Do the authors claim such accuracy in their base numbers? One doubts. For tas, again except for 1 station, essentially all RMSE lie below 1k. The authors expect us to believe with their tools they can distinguish products at 2 mm/day and 1k? Remarkable if true but they give us no evidence. A low correlation error (RMSE) between two products of assumed 'perfection' but almost certainly with high inherent fundamental uncertainties seems of little relevance?*

**Reply:** We agree that we have focused on the basins where a reduction of the RMSE was visible. In the revised manuscript, we will also cover the cases where the bias-correction has no or a negative impact. However, there seems to be a general misunderstanding with respect to the quantities that are shown and analyzed. Figure 2 and 3 are based on **basin-averages** from ERA5-Land, SEAS5 and SEAS5-BCSD and do not show a comparison between station-based observations and forecasts. That being said, averaging across a basin (or domain, area, etc.) acts as a kind of "filter" (similar to computing monthly from daily data) and differences between such averages are, by nature, smaller than comparing e.g., station-based observations with pixel-based data from a model with a spatial resolution of 10km and more.

We do think that differences of **basin-averaged monthly averages** at the 2mm/day and 1K level are worth to mention. As a quantitative example, for the Catamayo-Chira-basin, the RMSE during the peak of the rainy season (February/March) is reduced by 2mm/day (or 60mm/month). On average, seasonal precipitation between January and April is around 1100mm/season (or 275mm/month or around 10mm/day). Hence, a RMSE-reduction of 2mm/day refers to around 20% of the total precipitation during the peak rainy season and we think that this is a quite substantial improvement. This example also puts the RMSE-values across the other basins into a quantitative context. For the Karun, we have (even after BCSD) RMSE-values of around 2mm/day and this refers to almost 40% of the average total precipitation during the four peak months of the rainy season (3,9mm/day). Similarly, the climatological ranges of basin-averaged temperatures are 5K (Saõ Franciso), 30K (Karun), 7K (Blue Nile and Tekeze-Atbara), and 2K (Catamayo-Chira). So, depending on the region, RMSEs of mean monthly precipitation and temperature forecasts often have magnitudes of 0-4 mm/day and 0-2K (or 20 – 40% with respect to the long-term mean). These values are also in-line with similar studies (see e.g., Gerlitz et al. 2016 or Zebaze et al. 2019).

From a conceptual point of view, outliers or larger errors get more weight in the calculation of the RMSE compared to the bias. If there are certain months with large differences between the forecasts and the reference (which could even receive some correction in the "wrong" direction), the RMSE after bias-correction can remain unchanged or (in some cases) even worse.

*Page 10: Reader needs to jump from Fig 3 in 4.1 to Fig 6 in 4.2 then back to Fig 4 in 4.3. Reason for this hopping around? Hopping will disappear once Figures take their appropriate place in final document but then sequence will look wrong?*

**Reply:** This is true. We will re-arrange the sequence of Figures in the revised manuscript.

**Changes:** We have changed the order of Figures.

*Page 10 Section 4.2 resolution: no uncertainties here? These are average sums of 4-month periods from 25 to 51 ensemble runs over 35 years. They must have SD, 95CI, etc? Almost every number and result across the manuscript has substantial uncertainty ranges but authors treat everything as exact?*

**Reply:** The main goal of the presented approach is the correction of biases. In order to show the performance of the chosen method, we need to analyze the long-term biases between the reference information and the forecasts. And this is what we've done in section 4.2. Nevertheless, in order to focus more on the spread or uncertainty of the forecasts, we'll also include some discussion about the standard deviations of the reference data as well as the raw and corrected forecasts (see Figure 1).

**Changes:** In the revised version, we have included additional maps which show the standard deviation of seasonal precipitation. This helps to see if the variability of the forecasts over time is also corrected towards the reference information. In fact, particularly over the mountainous regions, the raw forecasts predicted a too low variability while SEAS5 BCSD agrees much better with ERA5-Land (page 11, lines 21-26).

[Figure]

*Figure 1: Total seasonal precipitation (left) and standard deviation of precipitation (right) from SEAS5 raw, SEAS5 BCSD and ERA5-Land for the four main months of the rainy seasons, over the period 1981 to 2016. This figure will replace the "old" Figure 6 as it shows that also the precipitation dynamics of SEAS5 BCSD agree better with the ERA5-Land-reference.*

*Page 10 section 4.3 lead-time: without ranges or uncertainties, reader has no basis to accept any of these supposed differences or patterns.*

**Reply:** The reviewer is absolutely right in that all these maps are based on an average across a long period of time and 25 ensemble members and, hence, can be provided with some statistical quantities (standard deviation, etc.). However, in this plot and the corresponding section, we would like to focus on the model drift of the whole SEAS5-forecasting system and why it is important to remove this effect.

*Page 10 line 20: weather patterns may shift but locations do not shift, southward or any other direction*

**Reply:** We will re-phrase the respective part to "A shift of higher temperatures and higher radiations with increasing lead times towards south. "

**Changes:** The part has been re-phrased (page 11, lines 31-32)

*Page 11 line 3: reader needs to go from Fig 4 in previous paragraph now to Fig 7. Consider a more helpful and logical sequence??*

**Reply:** This is truly confusing. We will re-arrange the sequence of Figures in the revised manuscript.

**Changes:** We have exchanged Figure 4 and 5. Now, the ordering should be correct.

*Page 11 line 5: reader now moves from geographic codes KA or CC back to domain codes D03. Why? Confusing!*

**Reply:** We are evaluating the forecasts over domains (D01 to D04) and river basins (KA, SF, BN, TA, CC). This is why we sometimes switch between geographic and domain codes. We will make this clearer in the revised manuscript (see also our reply to the comment for *Page 1 line 19*).

*Page 11 section 4.5 overall skill: many readers will know these skill scores but will usually have seen them expressed as a range. This reader has no confidence in an absolute CRPSS of 0.4 but might accept a range from 0.3 to 0.5? Again, authors treat their results as absolute when in fact they contain substantial uncertainty!*

**Reply:** We agree that we treat the CRPSS-values as "absolute" results but it depends on the application if a range of values or a simple mean makes more sense. By the very nature of the CRPSS, it requires an ensemble forecast and, hence, also takes the spread and statistical ensemble distribution into account. In order to analyze the performance of a forecasting system, we need to average across one or multiple dimensions (usually time) just like any other performance metric (like correlation, NSE, RMSE, etc.). In our case, we're showing the median over 36 years, which is fully consistent with many other studies (Yuan et al. 2015, Dutra et al. 2013, Lin et al. 2020, Dirkson et al. 2019). On the other hand, Steiger et al. (2018) show a boxplot of CRPSS-values (as requested by the reviewer), where the spread is computed across all global individual, pixel-based CRPSS-values. But the individual values, which go into the boxplot, are computed in exactly the same way as in our Figure 8. Similarly, Arnal et al. (2018), Woldemeskel et al. (2018) or Bazile et al. (2017) show boxplots and ranges of CRPSS values but such analyses are based on an ensemble across many regions or river basins (and NOT time). In such applications, it makes certainly sense to look at the range of CRPSS-values as the authors compute this range from individual CRPSS-values across many pixels, regions, domains, basins, etc. In our study, we focus on individual basins and show how the CRPSS varies between different lead-times and forecasted moths. Adding some uncertainty bounds to our CRPSS-analysis would be actually a subsequent step (e.g. across all basins, which would be similar to the workflow in e.g. Bazile et al. 2017). However, we do not think that computing uncertainty bounds from only five values gives any additional insights.

*Page 11 section 5 Discussion: helpful discussion of regional factors follows, intended apparently as justification for why corrected products seem occasionally but not consistently to outperform original forecasts. Very real regional challenges, no doubt. But if the original forecast products lacked sufficient skill when confronted by meteorological and topographic details of each basin, bias correction to higher resolution will not remove that fundamental detail-driven uncertainty? It may raise skill scores but still miss key local details. E.g. it will continue to show high fundamental uncertainty! Vis "spatial and temporal inconsistencies in the forecasted spatial extent and intensity" (Page 12 line 7) of precip, of temperature, of clouds, etc. represent the real-world uncertainty not included and certainly not overcome! The authors themselves make this point (Page 12 line 13) that for basins with skill score improvements of 0 and no differences in RMSE, fundamental uncertainty has defeated their good efforts!*

**Reply:** First of all, we were indeed able to demonstrate that our forecasts outperform the raw forecasts: Figure 2 shows a reduction of bias (which is the main impact of a bias-correction) across all basins and all variables and Figure 8 shows positive CPRSS-values across the majority of variables, forecasted and lead months. Therefore, we think that the term *occasionally* is not justified.

We of course agree that differences between forecasts and any regional reference (no matter if it is a merged or purely station-based product) can be attributed to fundamental, detail-driven uncertainty and the lack of local details. We can only improve the forecasts by bias-correction when they already provide a certain degree of skill, i.e., when the raw forecasts are already able to represent general circulation patterns and processes. The bias correction and spatial disaggregation are then able to introduce smaller-scale details (and implicitly smaller-scale processes) through the reference data. An explicit treatment of smaller-scale details and processes would require dynamical downscaling using a complex atmospheric model to improve the spatial resolution of atmospheric variables. From a more technically point of view (and our own experience), doing such dynamical downscaling experiments for an ensemble forecasting system for a period of almost 40 years and 25 and more ensemble members results requires tremendous computational resources, which is why approaches like BCSD and other statistical-empirical techniques gained more and more popularity. Regarding dynamical downscaling, several studies (e.g. even of ourselves: Klein et al. 2015 or Yang et al. 2021) further showed that the used parameterizations of small-scale processes in the models further introduce high uncertainties that can completely change the performance skill of the original data set.

Furthermore, the reviewer states that *fundamental uncertainty has defeated their* good *efforts!* Again, we think that this is a too pessimistic view as e.g., over the Sao Francisco Basin, where the raw forecasts for December, March and April were already quite good, we do not think that we were defeated by fundamental uncertainty but rather did not improve much upon the raw forecasts (which was already stated in our manuscript).

**Changes:** We have extended the review and discussion of „negative" results (page 11, lines 4-8; page 13, lines 6-7).

*Page 27 Figure 2: These are composite biases (areal sum of daily data) for source forecast vs ERA5-Land reanalysis? The colours - almost impossible to distinguish even in the label) represent different lead times from 0 to 11 months? Or are these monthly averages? Not clear. After working extensively similar Fig 3, I still find these graphics difficult to read and interpret.*

**Reply:** We fully agree and will change the Figure in a revised manuscript.

**Changes:** In the revised version, we have removed the colours for the individual months. Initially, we wanted to show the Biases and RMSEs from each single month and how they can differ. However, after carefully looking at the figure and what it should tell the reader, we think that it is much better to just show the raw and bias-corrected forecasts as grey and black, respectively. If a reader is still interested in the Biases or RMSEs from individual months, we have also added some x-ticks which allow for a better identification of single months.

*At this point this reader largely 'gave up'. The data description for DKRZ seems easy to use and very helpful. Authors have provided useful guidance to static products and how to find updates. Generally ESSD does not allow: 'contact the author' (Page 15 line 23). Appendices provide useful documentation on BC, on error calculations, and on skill scores. Overall the authors have provided useful information. Their approach however still seems orthogonal to the intent of ESSD.*

**Reply:** Thank you very much for these generally positive final words. The reason why we've included the contact information is that the DKRZ hosts the "hindcast" product from 1981 to 2019, while the operational forecasts are only available via the KIT Campus Alpin DataServer. But if it is necessary, we can of course remove this information.

**References:**

Andrade, C. W. L.; Montenegro, S. M. G. L.; Montenegro, A. A. A.; Lima, J. R. d. S.; Srinivasan, R.; Jones, C. A. (2020): Climate change impact assessment on water resources under RCP scenarios: A case study in Mundaú River Basin, Northeastern Brazil, *Int J Climatol*, doi: [10.1002/joc.6751](10.1002/joc.6751)

Arnal, L.; Cloke, H. L.; Stephens, E.; Wetterhall, F.; Prudhomme, C.; Neumann, J.; Krzeminski, B.; Pappenberger, F. (2018): Skilful seasonal forecasts of streamflow over Europe? *Hydrol. Earth Syst. Sci.*, doi: 10.5194/hess-22-2057-2018

Basheer, M.; Wheeler, K. G.; Elagib, N. A.; Etichia, M.; Zagona, E. A.; Abdo, G. M.; Harou, J. J. (2020). Filling Africa's largest hydropower dam should consider engineering realities. *One Earth*, doi: 10.1016/j.oneear.2020.08.015

Bazile, R.; Boucher, M.-A.; Perreault, L.; Leconte, R. (2017): Verification of ECMWF System 4 for seasonal hydrological forecasting in a northern climate. *Hydrol. Earth Syst. Sci.*, doi: 10.5194/hess-21-5747-2017

Dirkson, A.; Denis, B.; Merryfield, W. J. (2019): A Multimodel Approach for Improving Seasonal Probabilistic Forecasts of Regional Arctic Sea Ice. *Geophysical Research Letters*, doi: 10.1029/2019GL083831

Dutra, E.; Di Giuseppe, F.; Wetterhall, F.; Pappenberger, F. (2013): Seasonal forecasts of droughts in African basins using the Standardized Precipitation Index. *Hydrol. Earth Syst. Sci.*, doi: 10.5194/hess-17-2359-2013

ECMWF (2019): ERA5-Land hourly data from 1981 to present, Tech. rep., ECMWF, doi: 10.24381/cds.e2161bac

Fischer, E. M. and Knutti, R. (2014): Detection of spatially aggregated changes in temperature and precipitation extremes, *Geophys. Res. Lett.*, doi:10.1002/2013GL058499.

Gerlitz, L.; Vorogushyn, S.; Gafurov, A. (2012): Climate informed seasonal forecast of water availability in Central Asia: State-of-the-art and decision making context, *Water Security*, doi: 10.1016/j.wasec.2020.100061

Gerlitz, L.; Vorogushyn, S.; Apel, H.; Gafurov, A.; Unger-Shayesteh, K.; Merz, B. (2016): A statistically based seasonal precipitation forecast model with automatic predictor selection and its application to central and south Asia, *Hydrol. Earth Syst. Sci.,* doi: 10.5194/hess-20-4605-2016, 2016.

Gleixner, S.; Demissie, T.; Diro, G.T. (2020): Did ERA5 Improve Temperature and Precipitation Reanalysis over East Africa? *Atmosphere*, doi: 10.3390/atmos11090996

Heinzeller, D.; Dieng, D.; Smiatek, G.; Olusegun, C.; Klein, C.; Hamann, I.; Salack, S.; Bliefernicht, J.; Kunstmann, H. (2018): The WASCAL high-resolution regional climate simulation ensemble for West Africa: concept, dissemination and assessment, *Earth Syst. Sci. Data*, doi: 10.5194/essd-10-815-2018, 2018

Kidus, A. E. (2019): Long-term potential impact of Great Ethiopian Renaissance Dam (GERD) on the downstream eastern Nile High Aswan Dam (HAD), *Sustainable Water Resources Management*, doi: 10.1007/s40899-019-00351-0

Klein, C.; Heinzeller, D.; Bliefernicht, J. et al. (2015): Variability of West African monsoon patterns generated by a WRF multi-physics ensemble. *Climate Dynamics,* doi: 10.1007/s00382-015-2505-5

Lin, H.; Merryfield, W. J.; Muncaster, R.; Smith, G. C.; Markovic, M.; Dupont, F.; Roy, F.; Lemieux, J.-F.; Dirkson, A.; Kharin, V. V.; Lee, W.-S.; Charron, M.; Erfani, A. (2020): The Canadian Seasonal to Interannual Prediction System Version 2 (CanSIPSv2). *Weather and Forecasting,* doi: 10.1175/WAF-D-19-0259.1

Lorenz, C. and Kunstmann, H. (2012): The Hydrological Cycle in Three State-of-the-Art Reanalyses: Intercomparison and Performance Analysis, *Journal of Hydrometeorology*, doi: 10.1175/JHM-D-11-088.1, 2012.

Lorenz, C.; Kunstmann, H.; Devaraju, B.; Tourian, M. J.; Sneeuw, N.; Riegger, J.; Kunstmann, H. (2014): Large-scale runoff from landmasses: a global assessment of the closure of the hydrological and atmospheric water balances, *Journal of Hydrometeorology*, doi: 10.1175/JHM-D-13-0157.1, 2014.

Lorenz, C.; Tourian, M. J.; Devaraju, B.; Sneeuw, N.; Kunstmann, H. (2015): Basin-scale runoff prediction: An Ensemble Kalman Filter framework based on global hydrometeorological data sets, *Water Resour. Res.*, doi:10.1002/2014WR016794.

Marengo, J.A.; Chou, S.C.; Kay, G. et al. (2012): Development of regional future climate change scenarios in South America using the Eta CPTEC/HadCM3 climate change projections: climatology and regional analyses for the Amazon, São Francisco and the Paraná River basins. *Climate Dynamics,* doi: 10.1007/s00382-011-1155-5

Muñoz-Sabater, J.; Dutra, E.,Balsamo, G.; Boussetta, S.; Zsoter, E.; Albergel, C.; Agusti-Panareda, A. (2018): ERA5-Land: an improved version of the ERA5 reanalysis land component, *2nd International Surface Working Group (ISWG) & 8th Land Surface Analysis Satellite Application Facility (LSA-SAF)s Workshop*, https://confluence.ecmwf.int/download/attachments/50045219/Munoz_ILSWG_LSAF_Lisbon_2018_final.pdf?version=1&modificationDate=1531932262479&api=v2#

Muñoz Sabater, J.; Dutra, E.; Balsamo G. et al. (2017): ERA5-Land. A New state-of-the-art Global Land Surface Reanalysis Dataset, *31st Conference of Hydrology / 2017 American Meteorological Society (AMS) annual Meeting*, https://confluence.ecmwf.int/download/attachments/50045219/Munoz_AMS_Jan17_v2.pptx.pdf?version=1&modificationDate=1489419542877&api=v2

Patt, A. and Gwata, C. (2002): Effective seasonal climate forecast applications: examining constraints for subsistence farmers in Zimbabwe, *Global Environmental Change*, doi: 10.1016/S0959-3780(02)00013-4

Shukla, P. R., et al. (2019): IPCC, 2019: Climate Change and Land: an IPCC special report on climate change, desertification, land degradation, sustainable land management, food security, and greenhouse gas fluxes in terrestrial ecosystems, *in press*

Siegmund, J.; Bliefernicht, J.; Laux, P.; Kunstmann, H. (2015): Toward a seasonal precipitation prediction system for West Africa: Performance of CFSv2 and high-resolution dynamical downscaling. *J. Geophys. Res. Atmos.*, doi: 10.1002/2014JD022692

Sneeuw, N.; Lorenz, C.; Devaraju, B. et al. (2014): Estimating Runoff Using Hydro-Geodetic Approaches. *Surv Geophys.,* doi: *10.1007/s10712-014-9300-4*

Steiger, N. J.; Smerdon, J. E.; Cook, E. R.; Cook, B. I. (2018): A reconstruction of global hydroclimate and dynamical variables over the Common Era. *Scientific Data*, doi: 10.1038/sdata.2018.86

Tall, A.; Mason, S. J.; van Aalst, M.; Suarez, P.; Ait-Chellouche, Y.; Diallo, A. A.; Braman, L. (2012): Using Seasonal Climate Forecasts to Guide Disaster Management: The Red Cross Experience during the 2008 West Africa Floods, *International Journal of Geophysics*, doi: 10.1155/2012/986016, 2012

Tarek, M.; Brissette, F. P.; Arsenault, R. (2020): Evaluation of the ERA5 reanalysis as a potential reference dataset for hydrological modelling over North America, *Hydrol. Earth Syst. Sci.*, doi: 10.5194/hess-24-2527-2020, 2020.

Torres R.R.; Lapola D.M.; Gamarra N.L.R. (2017): Future Climate Change in the Caatinga. In: Silva J.M.C., Leal I.R., Tabarelli M. (eds) Caatinga. Springer, Cham. doi: 10.1007/978-3-319-68339-3_15

Touma, D.; Ashfaq, M.; Nayak, M. A.; Kao, S. C.; Diffenbaugh, N. S. (2015): A multi-model and multi-index evaluation of drought characteristics in the 21st century. *Journal of Hydrology*, doi: 10.1016/j.jhydrol.2014.12.011

Vladimirov E.; Dimitrova R.; Danchovski V. (2020): Impact of Data Assimilation on Short-Term Precipitation Forecasts Using WRF-ARW Model. In: Lirkov I., Margenov S. (eds) *Large-Scale Scientific Computing*, doi: 10.1007/978-3-030-41032-2_30

Wheeler, K. G.; Jeuland, M.; Hall, J. W.; Zagona, E.; Whittington, D. (2020): Understanding and managing new risks on the Nile with the Grand Ethiopian Renaissance Dam, Nature Communications, doi: 10.1038/s41467-020-19089-x

Woldemeskel, F.; McInerney, D.; Lerat, J.; Thyer, M.; Kavetski, D.; Shin, D.; Tuteja, N.; Kuczera, G. (2018): Evaluating post-processing approaches for monthly and seasonal streamflow forecasts. *Hydrology and Earth System Sciences*, doi: 10.5194/hess-22-6257-2018

Yang, Q. et al. (2021): Performance of the WRF model in simulating intense precipitation events over the Hanjiang River Basin, China - A multi-physics ensemble approach, *Atmospheric Research*, doi: 10.1016/j.atmosres.2020.105206.

Yuan, X.; Roundy, J. K.; Wood, E. F.; Sheffield, J. (2015): Seasonal Forecasting of Global Hydrologic Extremes: System Development and Evaluation over GEWEX Basins. *Bulletin of the American Meteorological Society*, doi: 10.1175/BAMS-D-14-00003.1

Zebaze, S.; Jain, S.; Salunke, P.; Shafiq, S.; Mishra, S. K. (2019): Assessment of CMIP5 multimodel mean for the historical climate of Africa. *Atmos Sci Lett*., doi: 10.1002/asl.926